# Efficient construction of the Feynman-Vernon influence functional as matrix product states

Chu Guo[1⋆] and Ruofan Chen[2†]

**1** Key Laboratory of Low-Dimensional Quantum Structures and Quantum Control of Ministry of Education, Department of Physics and Synergetic Innovation Center for Quantum Effects and Applications, Hunan Normal University, Changsha 410081, China
**2** College of Physics and Electronic Engineering, and Center for Computational Sciences, Sichuan Normal University, Chengdu 610068, China

⋆ guochu604b@gmail.com , † physcrf@sicnu.edu.cn

## Abstract

The time-evolving matrix product operator (TEMPO) method has become a very competitive numerical method for studying the real-time dynamics of quantum impurity problems. For small impurities, the most challenging calculation in TEMPO is to construct the matrix product state representation of the Feynman-Vernon influence functional. In this work we propose an efficient method for this task, which exploits the time-translationally invariant property of the influence functional. The required number of matrix product state multiplication in our method is almost independent of the total evolution time, as compared to the method originally used in TEMPO which requires a linearly scaling number of multiplications. The accuracy and efficiency of this method are demonstrated for the Toulouse model and the single impurity Anderson model.

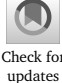

# 1  Introduction

A prototypical model for studying non-Markovian open quantum effects is to consider a quantum system of a few energy levels, referred to as the impurity, that is immersed in a continuous noninteracting bath which consists of an infinite number of bosonic or fermionic modes. Frequently studied quantum impurity models (QIMs) include the spin-boson model [1] which contains a single two-level spin coupled to a bosonic bath, and the Anderson impurity model which contains a localized electron coupled to an electron bath [2]. The former is a paradigmatic quantum system which is used to study, e.g., quantum stochastic resonance [3], dissipative Landau-Zener transition [4] and quantum phase transition [5,6]. The latter is a fundamental model for studying strong correlated effects in condensed matter physics [7], and is also a building block for quantum embedding methods such as the dynamical mean field theory [8].

A number of methods have been developed to solve the real-time dynamics of QIMs beyond the Born-Markov approximation. These method include the real-time diagrammatic Monte Carlo [9–11] and its recent higher order extensions [11,12], the hierarchical equation of motion [13–15], the numerical renormalization group [16–19], and the matrix product states (MPS) based methods [20–30]. Among these methods, the MPS based methods are known to allow well-controlled approximation schemes and could often be efficiently implemented [31,32].

An emerging and rapidly developing MPS based method for QIMs is the time-evolving matrix product operator (TEMPO) method, first developed for bosonic QIMs [33–40]. Recently this method was extended to the fermionic case, referred to as the Grassmann TEMPO (GTEMPO) method as it deals with the Grassmann path integral (PI) [41–43]. The central idea of TEMPO is to directly construct an MPS representation of the augmented density tensor (ADT), defined as the integrand of the PI, which only contains the impurity degrees of freedom in the temporal domain. This is in comparison with the conventional wave-functional based MPS methods which explicitly represent the spatial impurity-bath wave function as an MPS and then evolve it in time [29,30,44–46]. Here we note another set of recent works [47–51] that also exploits the MPS representation of the Feynman-Vernon influence functional (IF) [52] (which will be referred to as the tensor network IF methods). Formalism-wise, these methods differ from GTEMPO in that in their numerical calculations the PI is converted into a fermionic operator expression in the Fock state basis, thus avoiding directly dealing with Grassmann variables (GVs). In the meantime, the algorithm design in GTEMPO could be more straightforward as it directly translates the Grassmann expression of the fermionic PI into MPS calculations. The advantage of the TEMPO and the tensor network IF methods compared to the wavefunctional based MPS methods is obvious: the bath degrees of freedom are exactly treated via the Feynman-Vernon IF. The only sources of error in TEMPO are the time discretization error and the MPS bond truncation error. TEMPO is also likely to be advantageous in terms of computational efficiency compared to the wave-functional based methods. In fact, the entanglement of the temporal MPS is closely related to the memory kept in the bath that is relevant for the impurity dynamics [34,53], which thus resembles those natural orbital methods that

select a few relevant modes from the bath [54–57], but in TEMPO the whole process is done exactly without having to explicitly deal with the bath.

For small impurities, such as a single spin or a single electron, the construction of the MPS representation of the Feynman-Vernon IF (referred to as the MPS-IF afterwards) is the dominant calculation in TEMPO. The approach originally taken in TEMPO [33], is to build up the MPS-IF by decomposing the IF into the product of many partial IFs, each written as an MPS of a small bond dimension and the total number of partial IFs scales linearly with the total evolution time. Another approach to build the MPS-IF, which is used in the tensor network IF methods [47, 48], is to map the IF into a Gaussian state and then construct it by applying an equivalent quantum circuit of local gate operations onto the vacuum state using the Fishman-White (FW) algorithm [58]. The depth of the quantum circuit has been shown to scale only logarithmically with the evolution time in certain cases. The partial IF approach has also been used in combination with the tensor network IF method, and is reported to have a similar accuracy with the FW algorithm [50]. In both approaches, the time-translationally invariant (TTI) property of the IF is explicitly destroyed.

In this work we propose an alternative approach to construct the MPS-IF which exploits the TTI property of the IF. Similar to the partial IF approach, we build the IF using a series of MPS multiplications. However, the total number of MPS multiplications required by our approach is almost independent of the total evolution time (this feature shares some similarity to the logarithmic scaling of the circuit depth in the FW algorithm). In our numerical examples on the noninteracting Toulouse model and the single impurity Anderson model (SIAM), we find that a small constant number, 5 concretely, of MPS multiplications is already enough to achieve a similar level of accuracy to the partial IF approach, but with a drastic speedup in computational efficiency. Our method could thus greatly accelerate the TEMPO method for the real-time dynamics of QIMs.

## 2 Method description

The method we propose to efficiently construct the MPS-IF works for both the bosonic (TEMPO) and fermionic (GTEMPO) QIMs. In this section, we first present in detail our method for the fermionic case (which is the harder case), where Grassmann MPS (GMPS) is used instead of a standard MPS to represent the IF for the convenience of dealing with Grassmann tensors (one could refer to Ref. [41] for the definition and operations of GMPS). Then we briefly show its bosonic version.

### 2.1 The path integral formalism

For briefness we use the SIAM to describe our method, but we note that our method can be directly applied to general QIMs as long as the Feynman-Vernon IF applies, e.g., the bath is noninteracting and is linearly coupled to the impurity. The Hamiltonian of the SIAM can be written as

$$\hat{H} = \left(\epsilon_d - \frac{1}{2}U\right)\sum_\sigma \hat{a}_\sigma^\dagger \hat{a}_\sigma + U\hat{a}_\uparrow^\dagger \hat{a}_\uparrow \hat{a}_\downarrow^\dagger \hat{a}_\downarrow + \sum_{k,\sigma}\epsilon_k \hat{c}_{k,\sigma}^\dagger \hat{c}_{k,\sigma} + \sum_{k,\sigma}\left(V_k \hat{a}_\sigma^\dagger \hat{c}_{k,\sigma} + \text{H.c.}\right), \quad (1)$$

where the first line contains the impurity Hamiltonian, with $\sigma \in \{\uparrow, \downarrow\}$ the electron spin, $\epsilon_d$ the on-site energy of the impurity and $U$ the Coulomb interaction, the second line contains the bath Hamiltonian and the coupling between the impurity and the bath, with $\epsilon_k$ the band energy and $V_k$ the coupling strength. We assume that the whole system evolves from the initial state:

$$\hat{\rho}(0) = \hat{\rho}_{\text{imp}}(0) \otimes \hat{\rho}_{\text{bath}}^{\text{th}}, \quad (2)$$

where $\hat{\rho}_{\text{imp}}(0)$ is some arbitrary impurity state and $\hat{\rho}_{\text{bath}}^{\text{th}}$ is the bath equilibrium state with inverse temperature $\beta$.

The impurity partition function at time $t$, defined as $Z_{\text{imp}}(t) = \text{Tr}\,\hat{\rho}(t)/\text{Tr}\,\hat{\rho}_{\text{bath}}^{\text{th}}$, can be written as a path integral [59, 60]:

$$
\begin{aligned}
Z_{\text{imp}}(t) &= \int \mathcal{D}[\bar{a}, a] \mathcal{A}[\bar{a}, a] \\
&:= \int \mathcal{D}[\bar{a}, a] \mathcal{K}[\bar{a}, a] \prod_{\sigma} \mathcal{I}_{\sigma}[\bar{a}_{\sigma}, a_{\sigma}],
\end{aligned}
\tag{3}
$$

where $\bar{a}_{\sigma} = \{\bar{a}_{\sigma}(\tau)\}$, $a_{\sigma} = \{a_{\sigma}(\tau)\}$ are Grassmann trajectories, and $\bar{a} = \{\bar{a}_{\uparrow}, \bar{a}_{\downarrow}\}, a = \{a_{\uparrow}, a_{\downarrow}\}$ for briefness. The measure $\mathcal{D}[\bar{a}, a] = \prod_{\sigma,\tau} d\bar{a}_{\sigma}(\tau) da_{\sigma}(\tau) e^{-\bar{a}_{\sigma}(\tau) a_{\sigma}(\tau)}$. The integrand of the PI, denoted as $\mathcal{A}$, is the augmented density tensor (ADT) in the temporal domain which is a Grassmann tensor (it is a standard tensor of complex numbers for bosonic PI) and contains all the information of the impurity dynamics. It should be noted that this path integral formalism only contains the impurity GVs in the temporal domain.

The bare impurity propagator $\mathcal{K}$ is

$$
\begin{aligned}
\mathcal{K}[\bar{a}, a] = \langle -a_N^+|e^{-i\hat{H}_{\text{imp}}\delta t}|a_{N-1}^+\rangle \cdots \langle a_2^+|e^{-i\hat{H}_{\text{imp}}\delta t}|a_1^+\rangle \times \langle a_1^+|\hat{\rho}_{\text{imp}}(0)|a_1^-\rangle \langle a_1^-|e^{i\hat{H}_{\text{imp}}\delta t}|a_2^-\rangle \times \cdots \\
\times \langle a_{N-1}^-|e^{i\hat{H}_{\text{imp}}\delta t}|a_N^-\rangle,
\end{aligned}
\tag{4}
$$

where $\hat{H}_{\text{imp}}$ is the impurity Hamiltonian. For the purpose of numerical calculations, the IF can be discretized using the QuaPI method [61, 62] with a time step size $\delta t$, which results in the following discrete expression [41] (see Appendix. A for details):

$$
\mathcal{I}_{\sigma} \approx e^{-\sum_{\zeta,\zeta'}\sum_{jk} \bar{a}_{\sigma,j}^{\zeta} \Delta_{j,k}^{\zeta\zeta'} a_{\sigma,k}^{\zeta'}}.
\tag{5}
$$

Here $\zeta, \zeta' = \pm$ denotes the forward and backward branches of the Keldysh contour, $1 \leq j, k \leq N$ ($N = t/\delta t$ is the total number of time steps) label the discrete time steps, $\Delta_{j,k}^{\zeta\zeta'}$ denotes the four hybridization matrices, $a_{\sigma,k}^{\zeta}$ and $\bar{a}_{\sigma,k}^{\zeta}$ denote the discrete GVs. Since there are 8 GVs, $a_{\uparrow\downarrow,k}^{\pm}$ and $\bar{a}_{\uparrow\downarrow,k}^{\pm}$, within each time step, the total number of GVs is $8N$. For a given $\beta$, the hybridization matrices are fully determined by the coupling strength function: $J(\omega) = \sum_k V_k^2 \delta(\omega - \omega_k)$.

## 2.2 The partial IF method

In TEMPO, one first builds $\mathcal{K}$ and each $\mathcal{I}_{\sigma}$ as an MPS, and then multiplies them together to obtain the ADT as an MPS. In our implementation we represent each GV as one site, therefore the MPS representations of $\mathcal{K}$ and $\mathcal{I}_{\sigma}$ all have $8N$ sites for the SIAM. Based on the ADT, one can easily calculate any multi-time correlations of the impurity. In GTEMPO, a zipup algorithm is introduced to build the ADT only on the fly which could often further reduce the computational cost. In both cases, the most computationally expensive task is to build $\mathcal{I}_{\sigma}$ as an MPS (as long as the impurity is small).

Before we introduce our method, we first briefly review the partial IF method used in GTEMPO, which breaks $\mathcal{I}_{\sigma}$ into the product of partial IFs as:

$$
\mathcal{I}_{\sigma} = \prod_{\zeta,j} \mathcal{I}_{\sigma}^{\zeta,j} := \prod_{\zeta,j} \left( e^{-\sum_{\zeta',k} \bar{a}_{\sigma,j}^{\zeta} \Delta_{j,k}^{\zeta\zeta'} a_{\sigma,k}^{\zeta'}} \right).
\tag{6}
$$

Here $\mathcal{I}_{\sigma}^{\zeta,j}$ denotes the partial IF for the $\zeta$ branch and $j$th time step. The decomposition in Eq.(6) is exact since the partial IFs commute with each other (more generally, any Grassmann

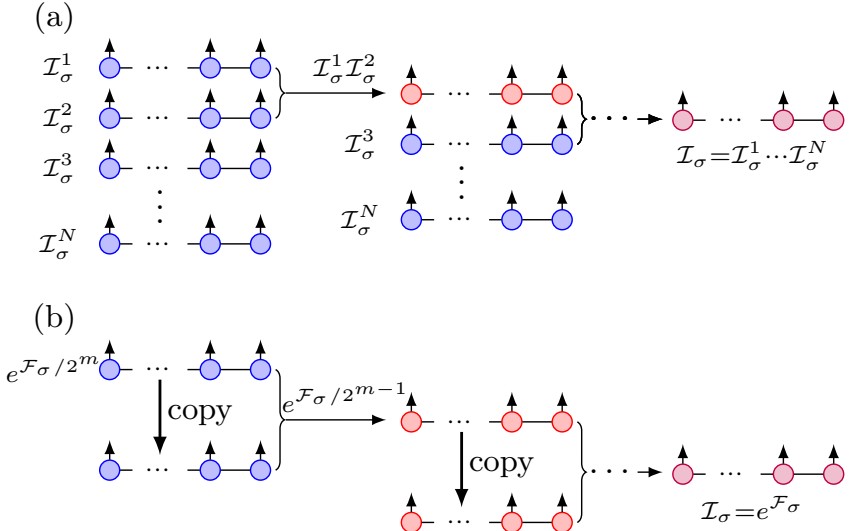

Figure 1: (a) The partial IF method to build the discretized $N$-step Feynman-Vernon influence functional as an MPS using the multiplications of $O(N)$ partial IFs. The branch indices of the partial IFs are suppressed for briefness. (b) The time-translationally invariant approach to build the MPS-IF, where $m$ GMPS multiplications is required and the result converges exponentially fast with $m$.

expression with an even number of GVs commutes with each other). In Ref. [41], a numerical algorithm is used to build each partial IF as a GMPS of a small bond dimension. Here we show that each partial IF can be exactly written as a GMPS of bond dimension 2, which is essentially because that the summand in its exponent shares the same GV $\bar{a}_{\sigma,j}^{\zeta}$ and one simply has

$$\mathcal{I}_\sigma^{\zeta,j} = 1 - \sum_{\zeta',k} \bar{a}_{\sigma,j}^{\zeta} \Delta_{j,k}^{\zeta\zeta'} a_{\sigma,k}^{\zeta'}. \tag{7}$$

Concretely, assuming that we use a *time-local ordering* of the GVs, where the GVs at different time steps are aligned in ascending order, and the GVs within the same time step $j$ are aligned as $a_{\sigma,j}^+ \bar{a}_{\sigma,j}^+ a_{\sigma,j}^- \bar{a}_{\sigma,j}^-$ [41,42], then the GMPS for $\mathcal{I}_\sigma^{\zeta,j}$ can be directly written as:

$$\mathcal{I}_\sigma^{\zeta,j} = \begin{bmatrix} 1 & \Delta_{j,1}^{\zeta\zeta'} a_{\sigma,1}^{\zeta'} \end{bmatrix} \cdots \begin{bmatrix} 1 & \Delta_{j,j}^{\zeta\zeta'} a_{\sigma,j}^{\zeta'} \\ 0 & 1 \end{bmatrix} \begin{bmatrix} 1 & \bar{a}_{\sigma,j}^{\zeta} \\ \bar{a}_{\sigma,j}^{\zeta} & 0 \end{bmatrix} \begin{bmatrix} 1 & 0 \\ -\Delta_{j,j+1}^{\zeta\zeta'} a_{\sigma,j+1}^{\zeta'} & 1 \end{bmatrix} \cdots \begin{bmatrix} 1 \\ -\Delta_{j,N}^{\zeta\zeta'} a_{\sigma,N}^{\zeta'} \end{bmatrix}, \tag{8}$$

where $\zeta'$ include both branches. This partial IF strategy is schematically shown in Fig. 1(a). Here we can also see another stark difference between GTEMPO and the wave-function based MPS methods: in GTEMPO the whole time evolution window $[0, t]$ is addressed simultaneously, while in the latter the evolution proceeds from time $t$ to $t + \delta t$ iteratively. Furthermore, in GTEMPO we first build the MPS-IF for the whole time interval $[0, t]$, which is then used for calculating any multi-time impurity correlations within this time interval.

## 2.3 The time-translationally invariant IF method

The partial IF method is generic for arbitrary hybridization matrix. However, this is an overkill since the hybridization matrix in Eq.(5) is not general: it has the crucial property of being time-translationally invariant, namely $\Delta_{j,k}^{\zeta\zeta'}$ can be written as a single-variate function of $j - k$ as

$$\Delta_{j,k}^{\zeta\zeta'} = \eta_{j-k}^{\zeta\zeta'}. \tag{9}$$

Denoting $\mathcal{F}_\sigma^{\zeta\zeta'} = -\sum_{jk} \bar{a}_{\sigma,j}^\zeta \Delta_{j,k}^{\zeta\zeta'} a_{\sigma,k}^{\zeta'}$ and $\mathcal{F}_\sigma = \sum_{\zeta,\zeta'} \mathcal{F}_\sigma^{\zeta,\zeta'}$, the above property allows us to efficiently write $\mathcal{F}_\sigma^{\zeta\zeta'}$ as a GMPS of a small bond dimension, similar to the strategy used in constructing long-range matrix product operators (MPOs) [63]. The details are shown in the following.

First we assume that $\eta^{\zeta\zeta'}$ (for fixed $\zeta$ and $\zeta'$) can be decomposed into the summation of exponential functions as

$$\eta_x^{\zeta\zeta'} \approx \sum_{l=1}^n \alpha_l \lambda_l^{|x|}, \tag{10}$$

where $\alpha_l$ and $\lambda_l$ are parameters to be determined, and $x$ can be both positive and negative. Finding the optimal $\alpha_l$ and $\lambda_l$ in Eq.(10) is an important and well-studied task in signal processing, which can be solved by the Prony algorithm [64] (also see Appendix. C). We denote the error occurred in this decomposition as

$$\varsigma_p = \sum_x \left( \eta_x^{\zeta\zeta'} - \sum_{l=1}^n \alpha_l \lambda_l^{|x|} \right)^2. \tag{11}$$

In practice, we will increase $n$ in the Prony algorithm until $\varsigma_p$ is less than a given threshold. Once the optimal $\alpha_l$ and $\lambda_l$ are found, $\mathcal{F}_\sigma^{\zeta\zeta'}$ can be built as an translationally invariant GMPS with bond dimension $2n+2$, the site tensor of which can be written as an upper-triangular (or equivalently lower-triangular) operator matrix:

$$\begin{bmatrix} 1 & \alpha_1 a_\sigma^{\zeta'} & \cdots & \alpha_n a_\sigma^{\zeta'} & -\bar{\alpha}_1 \bar{a}_\sigma^\zeta & \cdots & -\bar{\alpha}_n \bar{a}_\sigma^\zeta & \eta_0^{\zeta\zeta'} a_\sigma^{\zeta'} \bar{a}_\sigma^\zeta \\ 0 & \lambda_1 & \cdots & 0 & 0 & \cdots & 0 & \lambda_1 \bar{a}_\sigma^\zeta \\ \vdots & \vdots & \cdots & \vdots & \vdots & \cdots & \vdots & \vdots \\ 0 & 0 & \cdots & \lambda_n & 0 & \cdots & 0 & \lambda_n \bar{a}_\sigma^\zeta \\ 0 & 0 & \cdots & 0 & \bar{\lambda}_1 & \cdots & 0 & \bar{\lambda}_1 a_\sigma^{\zeta'} \\ \vdots & \vdots & \cdots & \vdots & \vdots & \cdots & \vdots & \vdots \\ 0 & 0 & \cdots & 0 & 0 & \cdots & \bar{\lambda}_n & \bar{\lambda}_n a_\sigma^{\zeta'} \\ 0 & 0 & \cdots & 0 & 0 & \cdots & 0 & 1 \end{bmatrix}, \tag{12}$$

where $\alpha_l$ and $\lambda_l$ correspond to the expansion of $\eta_x^{\zeta\zeta'}$ for $1 \le x \le N$ in Eq.(10), while $\bar{\alpha}_l$ and $\bar{\lambda}_l$ correspond to the expansion of $\eta_x^{\zeta\zeta'}$ for $-N \le x \le -1$. We have also neglected the time step indices of $\bar{a}$ and $a$ due to the time-translational invariance. Crucially, $n$ often scales very slowly with $N$ [65–67]. In practice, we find that for commonly used coupling strength functions, one could easily reach $\varsigma_p \le 10^{-5}$ with $n \le 20$.

Now that we have an efficient GMPS representation of each $\mathcal{F}_\sigma^{\zeta\zeta'}$, we could use any MPO-based time-evolving algorithms [68], such as the time-dependent variational principle (TDVP) [69], to construct $\mathcal{I}_\sigma = e^{\mathcal{F}_\sigma}$ as a GMPS. In fact one can easily transform back and forth between a GMPS and an MPO: an MPO can be converted into a GMPS by applying it onto the Grassmann vacuum, while a GMPS can be converted into an MPO by copying its physical indices, with the Grassmann anticommutation relations properly taking care of. However, brute-force application of these methods will in general require $O(1/\delta)$ MPO-MPS multiplications, if we choose $\delta$ as the step size ($\delta$ is a hyperparameter which has a completely different meaning from the $\delta t$ used for discretizing the IF). In the following we introduce an approach which only requires $O(\log(1/\delta))$ GMPS multiplications instead. Assuming that $\delta = 1/2^m$, then we can write

$$\mathcal{I}_\sigma = \left( e^{\mathcal{F}_\sigma/2^m} \right)^{2^m}. \tag{13}$$

For large enough $m$, one can first find an efficient first-order approximation of $e^{\mathcal{F}_\sigma^{\zeta\zeta'}/2^m}$ as a GMPS of bond dimension $2n+1$, using the $W^I$ method for example [63]:

$$
\begin{bmatrix}
1 + \delta\eta_0^{\zeta\zeta'} a_\sigma^{\zeta'} \bar{a}_\sigma^\zeta & \alpha_1' a_\sigma^{\zeta'} & \cdots & \alpha_n' a_\sigma^{\zeta'} & \bar{\alpha}_1' \bar{a}_\sigma^\zeta & \cdots & \bar{\alpha}_n' \bar{a}_\sigma^\zeta \\
\sqrt{\delta}\lambda_1 \bar{a}_\sigma^\zeta & \lambda_1 & \cdots & 0 & 0 & \cdots & 0 \\
\vdots & \vdots & \cdots & \vdots & \vdots & \cdots & \vdots \\
\sqrt{\delta}\lambda_n \bar{a}_\sigma^\zeta & 0 & \cdots & \lambda_n & 0 & \cdots & 0 \\
\sqrt{\delta}\bar{\lambda}_1 a_\sigma^{\zeta'} & 0 & \cdots & 0 & \bar{\lambda}_1 & \cdots & 0 \\
\vdots & \cdots & \vdots & \vdots & \cdots & \vdots & \vdots \\
\sqrt{\delta}\bar{\lambda}_n a_\sigma^{\zeta'} & 0 & \cdots & 0 & 0 & \cdots & \bar{\lambda}_n
\end{bmatrix}, \tag{14}
$$

with $\alpha_i' = \sqrt{\delta}\alpha_i$ and $\bar{\alpha}_i' = -\sqrt{\delta}\bar{\alpha}_i$. In our actual implementation we use the $W^{II}$ method which is a better first-order approximation than $W^I$. Once we have an efficient GMPS representation of each $e^{\mathcal{F}_\sigma^{\zeta\zeta'}/2^m}$, we can multiply these four GMPSs together (with MPS bond truncation) to obtain an efficient GMPS representation of $e^{\mathcal{F}_\sigma/2^m}$ (note that $\mathcal{F}_\sigma^{\zeta\zeta'}$ commutes with each other), then $\mathcal{I}_\sigma$ can be obtained by only $m$ GMPS multiplications: in the $i$th step one simply multiplies $e^{\mathcal{F}_\sigma/2^{m-i+1}}$ with itself. Crucially, $m$ is not directly dependent on the total evolution time $t$ and it is clear that the error occurred in the first-order approximation of $e^{\mathcal{F}_\sigma/2^m}$ will decrease exponentially fast with $m$ ($m$ could still be indirectly dependent on $t$ since the precision of the first-order approximation of each $e^{\mathcal{F}_\sigma^{\zeta\zeta'}/2^m}$ will be affected by the norm of the matrix $\mathcal{F}_\sigma^{\zeta\zeta'}$, but the latter will at most only increase polynomially with $N$). This TTI approach to construct the MPS-IF is schematically shown in Fig. 1(b).

To this end, we discuss the implementation-wise difference of the GMPS multiplication used in the partial IF and the TTI IF approaches. For the partial IF approach, one needs to multiply a GMPS of bond dimension 2 with an existing GMPS (of a large bond dimension $\chi$). This is done by simply performing GMPS multiplication [41] followed by the standard SVD compression [31]: one performs a left-to-right sweep to prepare a left-canonical MPS without any bond truncation, and then a right-to-left sweep to prepare a right-canonical MPS during which MPS bond truncation is performed (later we will discuss the MPS bond truncation strategy we have used). Since one of the GMPS involved in the multiplication has a very small bond dimension 2, the computational cost of this operation (multiplication followed by SVD compression) scales as $O(N\chi^3)$. For the TTI IF approach, one needs to multiply two same GMPS whose bond dimension could be close to $\chi$, if the SVD compression method is used for the bond truncation of the resulting GMPS, then the cost of the left-to-right sweep would scale as $O(N\chi^6)$ since no bond truncation is performed in this stage. A better MPS compression strategy in this scenario would be the variational compression technique [70]: one initializes a GMPS with a fixed bond dimension $\chi$ and iteratively minimizes the distance of it with the multiplication of the two GMPSs (the multiplication can be computed on the fly to reduce memory usage). The variational compression technique is used in our implementation of the TTI IF approach. Overall, the computational costs of the partial IF and the TTI IF approaches scale as $O(N^2\chi^3)$ and $O(N\chi^4)$ respectively. Therefore the TTI IF approach will be beneficial for large $N$ (assuming that $\chi$ will remain approximately a constant as $N$ grows, see Appendix. D for further discussions on the computational costs).

## 2.4  TTI approach to construct the MPS-IF for bosonic QIMs

Finally we briefly consider the bosonic case, for which we focus on the spin-boson model as an example. The Hamiltonian can be written as [1]:

$$
\hat{H} = \hat{H}_S + \hat{\sigma}_z \sum_k V_k(\hat{b}_k + \hat{b}_k^\dagger) + \sum_k \omega_k \hat{b}_k^\dagger \hat{b}_k, \tag{15}
$$

where $\hat{H}_S$ is the impurity spin Hamiltonian, $\hat{b}_k^\dagger$ and $\hat{b}_k$ are bosonic creation and annihilation operators. Here we also note that for bosonic impurity problems the TTI property of the IF has already been explored to speedup the construction of the MPS-IF [71, 72], however the strategies introduced in these works are very different from our approach.

After discretization using the QuaPI method, the bosonic IF has a similar discrete expression as Eq.(5) [40, 62]:

$$\mathcal{I} \approx e^{-\sum_{\zeta,\zeta'} \sum_{j,k} s_j^\zeta \Delta_{j,k}^{\zeta\zeta'} s_k^{\zeta'}}. \tag{16}$$

Here $s \in \{1, -1\}$ is a normal scalar instead of a Grassmann variable. Another difference of bosonic IF compared to Grassmann IF is that the conjugate variable of $s$ is the same as itself, thus the length of the resulting MPS representation is only half of the fermionic case. The partial IF approach to construct the MPS-IF can be done in parallel with Eq.(6), except that in the bosonic case Eq.(7) does not exactly hold and one may not be able to analytically write down each partial IF as an MPS of bond dimension 2. Nevertheless, an algorithm is introduced in Ref. [33] to numerically construct each partial IF as an MPS of a small bond dimension.

The TTI approach to construct the MPS-IF can be done by strictly following the fermionic case, but making the substitution of the GVs $a$ and $\bar{a}$ in Eqs.(12,14) by $\hat{\sigma}_z$. For example, the bosonic version of Eq.(12) is simply

$$\begin{bmatrix} 1 & \alpha_1 \hat{\sigma}_z^{\zeta'} & \cdots & \alpha_n \hat{\sigma}_z^{\zeta'} & \bar{\alpha}_1 \hat{\sigma}_z^\zeta & \cdots & \bar{\alpha}_n \hat{\sigma}_z^\zeta & \eta_0^{\zeta\zeta'} \hat{\sigma}_z^{\zeta'} \hat{\sigma}_z^\zeta \\ 0 & \lambda_1 & \cdots & 0 & 0 & \cdots & 0 & \lambda_1 \hat{\sigma}_z^\zeta \\ \vdots & \vdots & \cdots & \vdots & \vdots & \cdots & \vdots & \vdots \\ 0 & 0 & \cdots & \lambda_n & 0 & \cdots & 0 & \lambda_n \hat{\sigma}_z^\zeta \\ 0 & 0 & \cdots & 0 & \bar{\lambda}_1 & \cdots & 0 & \bar{\lambda}_1 \hat{\sigma}_z^{\zeta'} \\ \vdots & \vdots & \cdots & \vdots & \vdots & \cdots & \vdots & \vdots \\ 0 & 0 & \cdots & 0 & 0 & \cdots & \bar{\lambda}_n & \bar{\lambda}_n \hat{\sigma}_z^{\zeta'} \\ 0 & 0 & \cdots & 0 & 0 & \cdots & 0 & 1 \end{bmatrix}, \tag{17}$$

and similarly for Eq.(14), noticing that the minus sign is missing due to the bosonic commutation relation.

# 3 Numerical results

In this section we numerically demonstrate the accuracy and efficiency of the TTI approach to construct the MPS-IF, with comparisons to the analytical solutions and the partial IF approach. We will focus on the fermionic QIMs for our numerical experiments.

First of all, we discuss about the sources of numerical errors in the partial IF and the TTI IF approaches. In the partial IF approach, the only approximation made on top of the time discretization of the IF is the MPS bond truncation, which can be controlled either by setting a bond truncation tolerance $\varsigma$ (throwing away any singular values with relative weights smaller than $\varsigma$) as done in Ref. [41], or by setting a maximum bond dimension $\chi$ (keeping $\chi$ states with largest weights after bond truncation) as done in Ref. [48], or both. In all our numerical tests of both approaches to build the MPS-IF, we first use a small tolerance $\varsigma = 10^{-7}$ and then use $\chi$ as a hard limit for MPS bond truncation (from Refs. [41, 43], $\varsigma = 10^{-6}$ could already be accurate enough for the $\delta t$ and the coupling strength function we use in this work, therefore we are essentially using the second criterion $\chi$ for bond truncation). The accuracy of the partial IF approach with respect to the MPS bond truncation tolerance has been thoroughly studied in Refs. [41–43]. For the TTI IF approach, there are two additional sources of errors compared to the partial IF approach: the error occurred in the Prony algorithm characterized

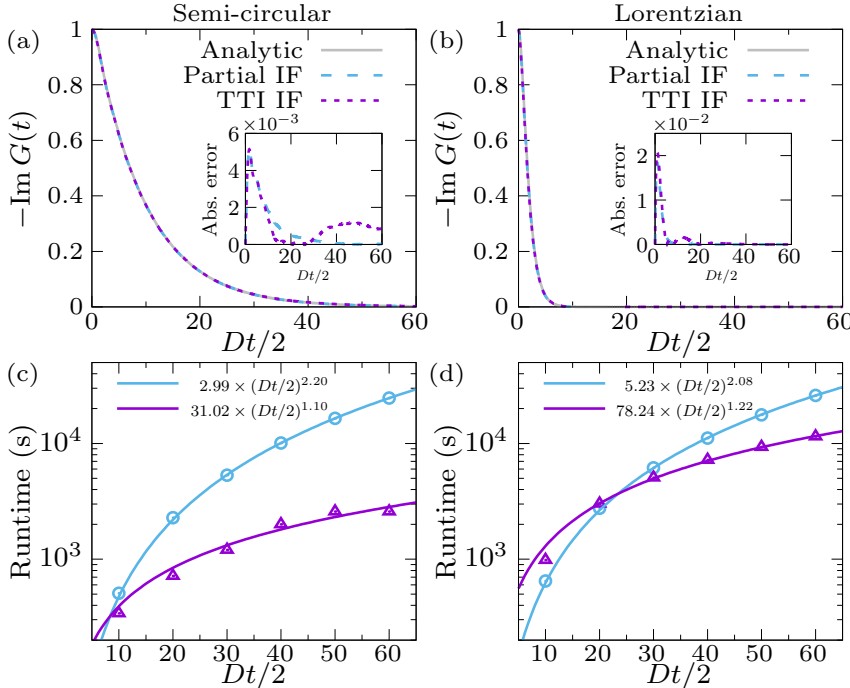

Figure 2: (a, b) Imaginary part of the retarded Green's function $G(t)$ versus time $t$ for (a) the semi-circular coupling strength function in Eq.(18) and (b) the Lorentzian coupling strength function in Eq.(19). The cyan and purple dashed lines are results for the partial IF and the TTI IF methods respectively. The insets show the absolute errors of both approaches compared to the analytical solutions. (c, d) The runtime scaling of both approaches for the two coupling strength functions used in (a, b). The cyan circle and purple triangle are results for the partial IF and the TTI IF respectively, the solid lines with the same colors are polynomial fittings for these two approaches. For these simulations we have used $\chi = 50$ for each MPS-IF, $D\delta t = 0.1$, $m = 5$ and $\varsigma_p = 10^{-5}$.

by $\varsigma_p$ and the discretization error of $\mathcal{I}_\sigma$ determined by $m$. In our numeric tests we will only consider the effects of these two additional sources of errors on the accuracy of the TTI IF approach.

## 3.1 The noninteracting ($U = 0$) case

The noninteracting Toulouse model [1,7] is a perfect test ground to access the accuracy and efficiency of our method, as this model is analytically solvable, and the way to construct the MPS-IF for this model is exactly the same as for more complicated impurity models: in the later cases one simply needs to construct more MPS-IFs. We will set $\epsilon_d = 0$ in the noninteracting case.

First, we use both the TTI IF method and the partial IF method to build the MPS-IF, where we set $\chi = 50$ for both methods and set $\varsigma_p = 10^{-5}$ (we require the error occurred in the Prony algorithm to be smaller than $\varsigma_p$), $m = 5$ for the TTI IF method, then we calculate the retarded Green's functions based on these two MPS-IFs respectively. To show the general performance of the TTI approach, we consider two very different coupling strength functions: (i) the semi-circular function

$$J_s(\omega) = \frac{\Gamma}{2\pi} D \sqrt{1 - (\omega/D)^2},\tag{18}$$

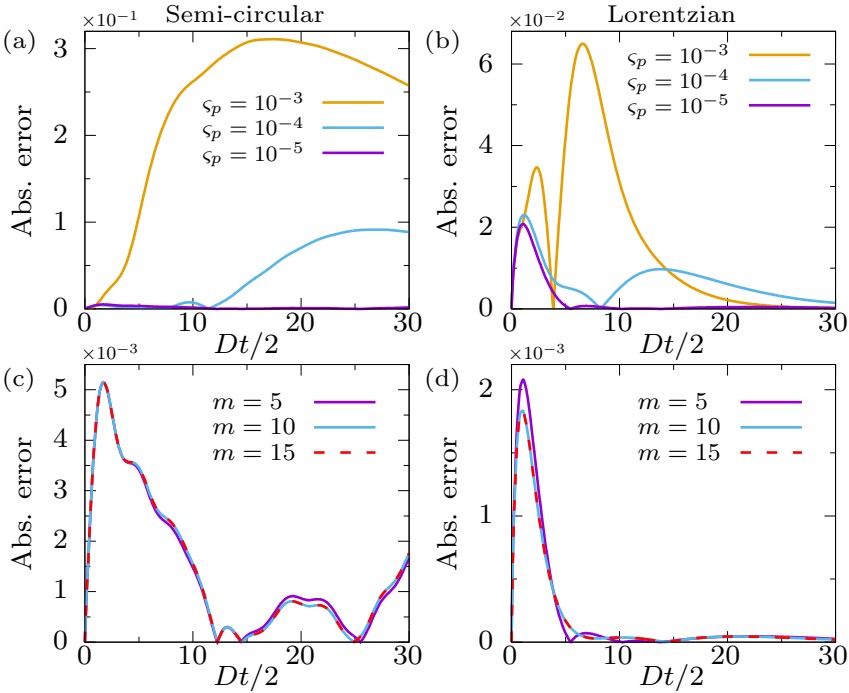

Figure 3: (a, c) The absolute errors between the retarded Green's function of the Toulouse model calculated by the TTI IF method and the analytical solutions for (a) different tolerance $\varsigma_p$ used in the Prony algorithm and (c) different values of $m$, for the semi-circular coupling strength function in Eq.(18). (b, d) The same plots as (a, c) but for the Lorentzian coupling strength function in Eq.(19). For these simulations we have used $\chi = 50$ for each MPS-IF and $D\delta t = 0.1$. The default values of the rest two hyperparameters, if not particularly specified, are $m = 5$ and $\varsigma_p = 10^{-5}$.

with $\Gamma = 0.1$, and (ii) the Lorentzian function

$$J_l(\omega) = \frac{1}{\pi} \frac{D}{\omega^2 + D^2}\,. \tag{19}$$

Here the coupling strengths in the two coupling strength functions are set to be very different on purpose to demonstrate the accuracy and universality of our method. In both cases we set $D = 2$ (we use $D/2$ as the unit) and set $D\delta t = 0.1$. We will also fix $D\beta = 10$ in all our numerical simulations. The results are plotted in Fig. 2 with $Dt = 120$ at most (with $N = 1200$). From Fig. 1(a, b), we can see that the absolute errors of the results calculated by the TTI IF method and by the partial IF method against analytical solutions are both of the order $10^{-3}$. From Fig. 1(c, d), we can see that the TTI IF method is significantly more efficient than the partial IF method: the former roughly scales as $t^1$ while the latter roughly scales as $t^2$. The $t^1$ scaling in the TTI IF method is because that the length of the underlying GMPS grows linearly with $t$ for a fixed $\delta t$, even though the number of GMPS multiplications is fixed as a constant.

In Fig. 3, we study the influence of the two hyperparameters: $\varsigma_p$ and $m$ on the accuracy of the TTI IF results. From Fig. 3(a, b), we can see that the error occurred in the Prony algorithm is crucial for the accuracy of the final results: we see drastic improvement of accuracy when decreasing $\varsigma_p$ from $10^{-3}$ to $10^{-5}$ for both coupling strength functions. In comparison, from Fig. 3(c, d), we can see that the improvement of accuracy by increasing $m$ is almost negligible, the results for a small $m = 5$ are already as accurate as those for $m = 15$.

To this end, we note that in our understanding of the scaling analysis of the computational costs for both methods, we have implicitly assumed that the required bond dimension $\chi$ is

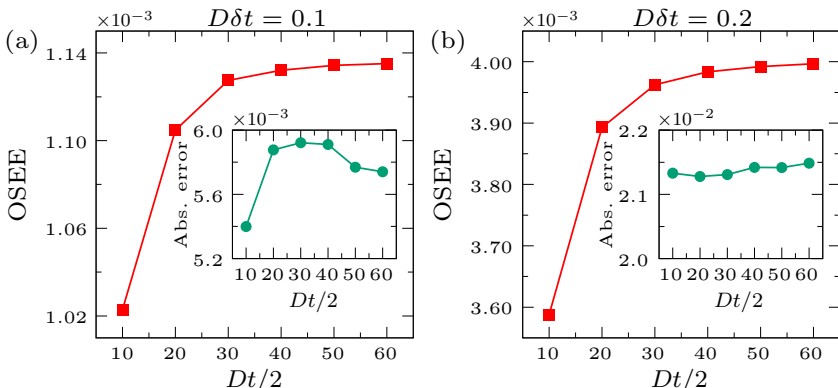

Figure 4: The maximum OSEE as a function of the total evolution time $t$ for the Toulouse model for (a) $D\delta t = 0.1$ and (b) $D\delta t = 0.2$. We have used $\chi = 100$ and the semi-circular coupling strength function in Eq.(18). We have also used the TTI IF method with $m = 5$ and $\varsigma_p = 10^{-5}$ to build the MPS-IF in these simulations. The insets in both panels show maximum absolute error against the analytical solutions of the retarded Green's function.

roughly a constant (which we fix to be 50 in above simulations) as $t$ increases. While currently there is no known theoretical guarantee that the IF can be efficiently represented as an MPS with a constant bond dimension, we can roughly understand the effectiveness of the MPS representation as follows: the exponent $\mathcal{F}_\sigma$ of the IF has a similar form to a spatially translationally invariant long-range quadratic Hamiltonian with coupling coefficients $\Delta_{j,k}^{\zeta\zeta'}$, which decays to zero as $|j - k| \to \infty$. Moreover, $\mathcal{I}_\sigma = e^{\mathcal{F}_\sigma}$ is similar to a high-temperate thermal state with Hamiltonian $-\mathcal{F}_\sigma$ and inverse temperature 1 (the constant 1 is crucial for the fast convergence of our simulation results against $m$ in Fig. 3). In practice, we find that we can generally obtain very accurate results with $\chi \leq 100$. To quantify the effectiveness of the MPS representation, we further plot in Fig. 4 the operator space entanglement entropy (OSEE), defined as the bipartition entanglement entropy of quantum operators or density operators [73] (the IF is similar to a density operator in the temporal domain [74]), as a function of the total evolution time $t$. We can see that the OSEE approximately saturates at $Dt/2 = 50$ for both $D\delta t = 0.1$ in Fig. 4(a) and $D\delta t = 0.2$ in Fig. 4(b), and the accuracy of our simulation becomes higher for smaller $\delta t$. The increase of OSEE for larger $\delta t$ also agrees with the observation in the bosonic case, where it is shown that a larger bond dimension is often required for larger $\delta t$ [38].

## 3.2 The single impurity Anderson model

As an application for a harder instance, we apply our method to study the steady state current of the single impurity Anderson model coupled to two baths. Our goal in this study is mainly to verify whether the existing finite-time calculations have reached steady state or not, by extending the evolution time with our new method. We use the same settings as used in Refs. [11, 41, 48], namely the two baths are both at zero temperature, with chemical potentials $\mu_1 = -\mu_2 = V/2$ and semi-circular coupling strength function in Eq.(18) with $\Gamma = 0.1$ (we use $\Gamma$ as the unit in this case). In Refs. [41, 48], the steady state particle current is computed by performing real-time evolution till $\Gamma t = 4.2$. In particular, the results in Ref. [41] calculated by the partial IF method have well converged against different $\delta t$ and MPS bond truncation tolerance, therefore the major remaining factor that may affect the quality of the obtained steady state current is the total evolution time (the whole system may have not reached its non-

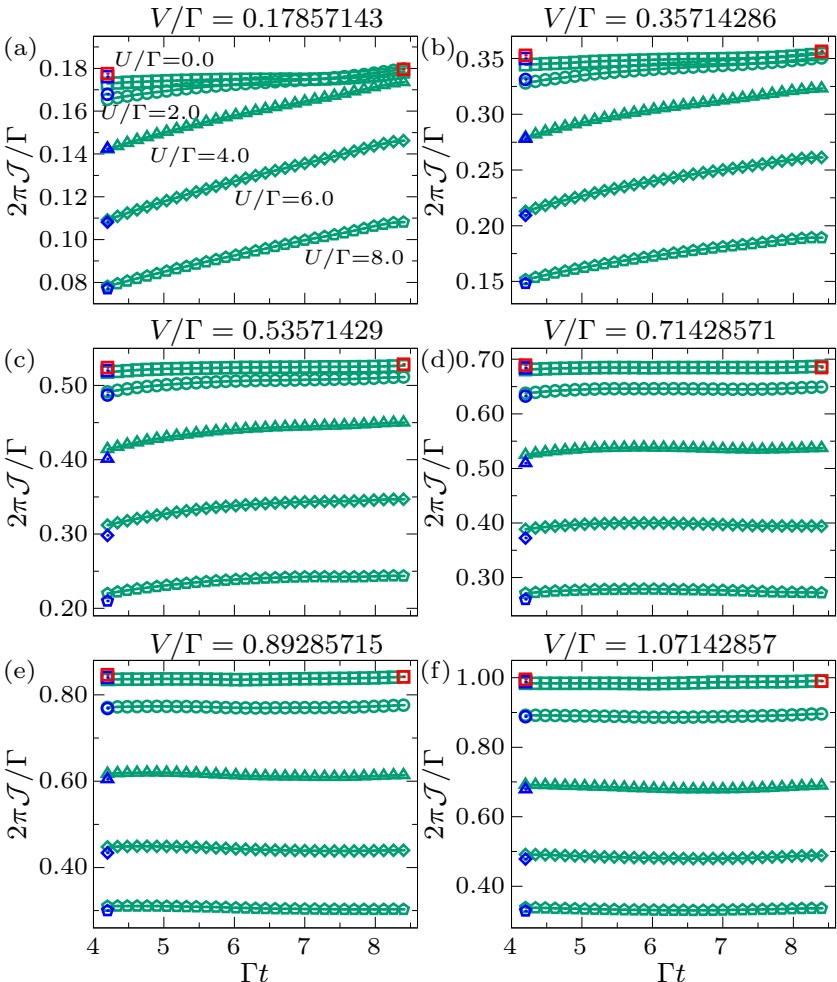

Figure 5: The symmetric particle current $\mathcal{J}$ as a function of time $t$ for different values of $V$ as indicated in the title of each panel. The green lines with markers are TTI IF results for $U/\Gamma = 0, 2, 4, 6, 8$ respectively. The red squares are the exact diagonalization results for $U = 0$. The blue markers are previous partial IF results from Ref. [41], calculated with $\varsigma = 10^{-7}$ (the bond dimensions of the MPS-IFs in these calculations are close to 160) and $\Gamma \delta t = 0.014$. For the TTI IF calculations we have used $\chi = 160$, $\Gamma \delta t = 0.014$, $m = 5$ and $\varsigma_p = 10^{-5}$.

equilibrium steady state yet with $\Gamma t = 4.2$). With the more efficient TTI IF method to construct the MPS-IF, we can reach longer evolution time. In this work, we thus evolve the system till $\Gamma t = 8.4$ (with $N = 600$) and check if the previous results have well converged to their steady state values. We denote the particle current from the $\nu$th bath with spin $\sigma$ into the impurity as $\mathcal{J}_\sigma^\nu$ (see Ref. [41] for the definition of particle current and the way to calculate it based on the obtained MPS-IFs). As in Refs. [41, 48], we calculate the symmetric particle current $\mathcal{J} = (\mathcal{J}_\uparrow^1 - \mathcal{J}_\uparrow^2)/2 = (\mathcal{J}_\downarrow^1 - \mathcal{J}_\downarrow^2)/2$. Here we also note that the major performance advantage of GTEMPO compared to the tensor network IF method is that the computational cost of GTEMPO is independent of the number of baths (as the baths are all integrated out in the Feynman-Vernon IF) [41], while the cost of the tensor network IF method scales exponentially with the number of baths [48].

In Fig. 5, we plot the symmetric particle current versus time $t$, with the starting point $\Gamma t = 4.2$ (which is the longest time that has been reached in previous studies [41, 48]), for $V$ from small to large. We focus on the regime of small chemical potential bias with $V/\Gamma \le 1.07$,

which is often harder to reach the steady state. We have used $\Gamma \delta t = 0.014$ and $\chi = 160$ for all these simulations. As comparison, we have shown the partial IF results taken from Ref. [41] (under the same $\delta t$) with the same markers but in cyan (only for the starting time). The results from exact diagonalization (ED) is also shown in the same markers but in red (for the starting and final times). For ED results we have discretized each bath into 8000 equal-distant frequencies and verified their convergence against bath discretization. We can see that the results calculated with our TTI IF method well matches with the previous partial IF results (the errors between them are within the first-order time discretization error). From Fig. 5(a, b, c) where $V/\Gamma < 0.54$, we can see that the particle currents has converged fairly well for $U = 0$, however, for $U > 0$ the particle current increases significantly (especially for $U/\Gamma > 2$) from $\Gamma t = 4.2$ to $\Gamma t = 8.4$, indicating that the whole system has not reached its steady state yet, and that the derivation from the steady state seems to be larger when $U/\Gamma$ increases from 0 to 4. In comparison, from Fig. 5(d,e,f) where $V/\Gamma > 0.7$, we can see that the particle currents have well converged for all values of $V$s and $U$s we have considered. Overall, the above results indicate that the system could reach its steady state more quickly for larger $V$ and smaller $U$.

## 4 Summary

In summary, we have proposed an efficient method to construct the MPS representation of the Feynman-Vernon influence functional, which is a central step in the TEMPO method and may also be applicable in the tensor network IF method. Our method exploits the time-translationally invariant property of the Feynman-Vernon IF for quantum impurity problems. Compared to the partial IF method originally used in TEMPO where the required number of MPS multiplications scales linearly with the total evolution time $t$, the number of MPS multiplications required in our method is almost independent of $t$. We demonstrate the accuracy and efficiency of our method in the noninteracting Toulouse model and the single impurity Anderson model with two baths, where we show that the TTI IF method can reach comparable accuracy with the existing partial IF method, but with a drastic speedup. Our method could thus significantly accelerate the TEMPO method for solving real-time dynamics of quantum impurity problems.

## Acknowledgments

**Funding information**   This work is supported by National Natural Science Foundation of China under Grant No. 12104328. C. G. is supported by the Open Research Fund from State Key Laboratory of High Performance Computing of China (Grant No. 202201-00).

## A   Quasi-adiabatic propagator scheme for hybridization functions

The hybridization function on the Keldysh contour $\Delta(\tau, \tau')$ is

$$\Delta(\tau, \tau') = \mathcal{P}_{\tau\tau'} \int d\omega J(\omega) D_\omega(\tau, \tau'), \tag{A.1}$$

where $D_\omega(\tau, \tau')$ is the free contour-ordered Green's function of the bath, defined as

$$D_\omega(\tau, \tau') = \langle T_{\mathcal{C}} \hat{c}_\omega(\tau) \hat{c}_\omega^\dagger(\tau') \rangle_0 . \tag{A.2}$$

Here $T_{\mathcal{C}}$ is the contour-ordered operator, and $P_{\tau\tau'} = 1$ if $\tau, \tau'$ are on the same Keldysh branch, and $-1$ otherwise. On the normal time axis, $D_\omega(\tau, \tau')$ is split into four blocks as

$$D_\omega(\tau, \tau') = \begin{bmatrix} D_\omega^{++}(t,t') & D_\omega^{+-}(t,t') \\ D_\omega^{-+}(t,t') & D_\omega^{--}(t,t') \end{bmatrix}, \tag{A.3}$$

where the explicit forms are

$$D_\omega^{++}(t,t') = \begin{cases} [1-n(\omega)]e^{-i\omega(t-t')}, & t \geq t', \\ -n(\omega)e^{-i\omega(t-t')}, & t < t', \end{cases} \tag{A.4}$$

$$D_\omega^{+-}(t,t') = -n(\omega)e^{-i\omega(t-t')}, \tag{A.5}$$

$$D_\omega^{-+}(t,t') = [1-n(\omega)]e^{-i\omega(t-t')}, \tag{A.6}$$

$$D_\omega^{--}(t,t') = \begin{cases} -n(\omega)e^{-i\omega(t-t')}, & t \geq t', \\ [1-n(\omega)]e^{-i\omega(t-t')}, & t < t'. \end{cases} \tag{A.7}$$

Here $n(\omega) = (e^{\beta\omega}+1)^{-1}$ is the Fermi-Dirac distribution function.

We split the trajectories $\bar{a}_\sigma^\pm(t), a_\sigma^\pm(t)$ into intervals of equal duration as $\bar{a}_{\sigma,j}^\pm, a_{\sigma,j}^\pm$ in $(j-\frac{1}{2})\delta t < t < (j+\frac{1}{2})\delta t$, then the hybridization function is discretized according to QuaPI scheme as

$$\Delta_{j,k}^{\zeta\zeta'} = \int d\omega J(\omega) \int_{(j-\frac{1}{2})\delta t}^{(j+\frac{1}{2})\delta t} dt \int_{(k-\frac{1}{2})\delta t}^{(k+\frac{1}{2})\delta t} dt' D_\omega(t,t'). \tag{A.8}$$

Note that $P_{\tau\tau'}$ vanishes in the above expression since it cancels the sign of $d\tau, d\tau'$ on the Keldysh contour. Then the explicit expressions of the discretized hybridization functions are

$$\Delta_{j,k}^{++} = \begin{cases} 2\int d\omega \frac{J(\omega)}{\omega^2}[1-n(\omega)]e^{-i\omega(j-k)\delta t}(1-\cos\omega\delta t), & j > k, \\ -2\int d\omega \frac{J(\omega)}{\omega^2}n(\omega)e^{-i\omega(j-k)\delta t}(1-\cos\omega\delta t), & j < k, \\ \int d\omega \frac{J(\omega)}{\omega^2}\{[1-n(\omega)][(1-i\omega\delta t)-e^{-i\omega\delta t}]-n(\omega)[(1+i\omega\delta t)-e^{i\omega\delta t}]\}, & j = k, \end{cases} \tag{A.9}$$

$$\Delta_{j,k}^{+-} = -2\int d\omega \frac{J(\omega)}{\omega^2}n(\omega)e^{-i\omega(j-k)\delta t}(1-\cos\omega\delta t), \tag{A.10}$$

$$\Delta_{j,k}^{-+} = 2\int d\omega \frac{J(\omega)}{\omega^2}[1-n(\omega)]e^{-i\omega(j-k)\delta t}(1-\cos\omega\delta t), \tag{A.11}$$

$$\Delta_{j,k}^{--} = \begin{cases} -2\int d\omega \frac{J(\omega)}{\omega^2}n(\omega)e^{-i\omega(j-k)\delta t}(1-\cos\omega\delta t), & j > k, \\ 2\int d\omega \frac{J(\omega)}{\omega^2}[1-n(\omega)]e^{-i\omega(j-k)\delta t}(1-\cos\omega\delta t), & j < k, \\ -\int d\omega \frac{J(\omega)}{\omega^2}\{n(\omega)[(1-i\omega\delta t)-e^{-i\omega\delta t}]-[1-n(\omega)][(1+i\omega\delta t)-e^{i\omega\delta t}]\}, & j = k. \end{cases} \tag{A.12}$$

## B  Multiplication of two Grassmann MPSs

In bosonic case, the discretized influence functional (IF) has the following form [33]:

$$\mathcal{I}[s_1^\pm, \dots, s_N^\pm] = e^{-\sum_{\zeta,\zeta'}\sum_{j,k}s_j^\zeta \Delta_{j,k}^{\zeta\zeta'}s_k^{\zeta'}}, \tag{B.1}$$

which can be expressed as the product of partial-IF as

$$\mathcal{I}[s_1^\pm,\ldots,s_N^\pm] = \prod_{j,\zeta} \mathcal{I}_{j,\zeta}[s_1^\pm,\ldots,s_N^\pm]$$
$$= \prod_{j,\zeta} e^{-\sum_{k,\zeta'} s_j^\zeta \Delta_{j,k}^{\zeta\zeta'} s_k^{\zeta'}}. \tag{B.2}$$

Here we can see that each partial IF is an ordinary tensor of $2N$ variables (e.g., of rank $2N$) from $s_1^\pm$ to $s_N^\pm$, and the IF can be obtained by element-wise product of the partial IFs. When the partial IFs are represented as MPSs, the element-wise product between two ordinary tensors is converted into the multiplication of two MPSs. As a result, for the multiplication of two MPSs, one should perform element-wise product for the physical indices and tensor product for the auxiliary indices.

In fermionic case, the influence functional has a similar form but the ordinary number $s$ is replaced by the Grassmann variable (GV) [60]. And we need to deal with the multiplication of Grassmann tensors (GT) in this case.

Assuming that for an algebra of GVs of $n$ components $\xi_1,\ldots,\xi_n$, we have two GTs

$$\mathcal{A} = \sum_i A^{i_n,\ldots,i_1} \xi_n^{i_n} \cdots \xi_1^{i_1}, \quad i_k = \{0,1\}, \tag{B.3}$$

and

$$\mathcal{B} = \sum_j A^{j_n,\ldots,j_1} \xi_n^{j_n} \cdots \xi_1^{j_1}, \quad j_k = \{0,1\}, \tag{B.4}$$

where the upper indices $i_k, j_k$ of GV $\xi_k$ are actual powers. The product of these two GTs gives

$$\tilde{\mathcal{C}} = \sum_{ij} A^{i_n,\ldots,i_1} B^{j_n,\ldots,j_1} \xi_n^{i_n} \cdots \xi_1^{i_1} \xi_n^{j_n} \cdots \xi_1^{j_1}. \tag{B.5}$$

We need to swap $\xi_n^{i_n},\ldots,\xi_1^{i_1}$ to the proper position, and each swapping would yields a sign according to the GV commutation rule. Due to this sign issue, the product of two GTs do not simply falls into the element-wise product of the coefficient tensors $A^{i_n,\ldots,i_1}$ and $B^{j_n,\ldots,j_1}$. After the rearrangement, we formally have $\tilde{\mathcal{C}}$ in the form

$$\tilde{\mathcal{C}} = \sum_{ij} \tilde{C}^{i_n j_n,\ldots,i_1 j_1} (\xi_n^{i_n} \xi_n^{j_n}) \cdots (\xi_1^{i_1} \xi_1^{j_1}), \tag{B.6}$$

where $\tilde{C}^{i_n j_n,\ldots,i_1 j_1}$ are obtained from the tensor product $A^{i_n,\ldots,i_1} B^{j_n,\ldots,j_1}$ and also by taking the swapping signs into consideration. The index $i_k, j_k$ can be merged by noticing the Grassmann multiplication relation (which corresponds to the element-wise product in the bosonic case)

$$\xi_k^{i_k=0} \xi_k^{j_k=0} = 1, \qquad \xi_k^{i_k=1} \xi_k^{j_k=1} = 0,$$
$$\xi_k^{i_k=1} \xi_k^{j_k=0} = \xi_k, \qquad \xi_k^{i_k=1} \xi_k^{j_k=0} = \xi_k. \tag{B.7}$$

After merging the index $i_k, j_k$, we finally obtain a GT $\mathcal{C}$ in the form

$$\mathcal{C} = \sum_i C^{i_n,\ldots,i_1} \xi_n^{i_n} \cdots \xi_1^{i_1}, \tag{B.8}$$

where the coefficient tensor $C^{i_n,\ldots,i_1}$ is obtained from $\tilde{C}^{i_n j_n,\ldots,i_1 j_1}$ by merging the index pairs $i_k, j_k$.

Now if we directly represent the coefficient tensor of the GT as an MPS, the resulting rule for MPS multiplication is to perform tensor product of each site tensor first, and then merge the physical indices using the rules in Eq.(B.7). In this case one should also take the global sign change into consideration, which would result in anonying global operations on the resulting MPS. This latter issue is perfected solved by the usage of Grassmann MPS (GMPS) which employs the $Z_2$ parity symmetry and reduces global sign changes into local ones [41].

# C The Prony algorithm

We want to approximate a discrete function $\eta_x$ in the form (here for simplicity we assume $x \geq 0$)

$$\eta_x \approx \sum_{l=1}^{n} \alpha_l \lambda_l^x. \tag{C.1}$$

This is a nonlinear problem even if the value of $n$ is known. The goal is to pick $2n$ samples $\eta_1, \ldots, \eta_{2n}$ to fit the $2n$ parameters $\alpha_1, \ldots, \alpha_n, \lambda_1, \ldots, \lambda_n$. The above expression gives $n$ equations which may be expressed in the matrix form as

$$\begin{pmatrix} \lambda_1^0 & \cdots & \lambda_n^0 \\ \vdots & \ddots & \vdots \\ \lambda_1^{n-1} & \cdots & \lambda_n^{n-1} \end{pmatrix} \begin{pmatrix} \alpha_1 \\ \vdots \\ \alpha_n \end{pmatrix} = \begin{pmatrix} \eta_1 \\ \vdots \\ \eta_n \end{pmatrix}. \tag{C.2}$$

Therefore if the $\lambda_l$s are determined, then we have a set of linear equations which can be easily solved to obtain the $\alpha_l$s.

The key of the Prony method is to recognize that Eq. (C.1) is the solution to some homogeneous linear constant-coefficient difference equation. In order to find such a difference equation, we define the characteristic polynomial $\phi(\lambda)$ as

$$\phi(\lambda) = \prod_{l=1}^{n} (\lambda - \lambda_l). \tag{C.3}$$

This polynomial has $\lambda_l$ as its roots and can be expanded into a power series as

$$\phi(\lambda) = \sum_{k=0}^{n} a_k \lambda^{n-k}, \tag{C.4}$$

where the coefficient $a_0 = 1$. Employing Eq. (C.1), we have

$$\begin{aligned} \sum_{k=0}^{n} a_k \eta_{p-k} &= \sum_{k=0}^{n} a_k \sum_{l=1}^{n} \alpha_l \lambda_l^{p-k-1} \\ &= \sum_{l=1}^{n} \alpha_l \lambda_l^{p-n-1} \sum_{k=0}^{n} a_k \lambda_l^{n-k} \\ &= \sum_{l=1}^{n} \alpha_l \lambda_l^{p-n-1} \phi(\lambda_l) = 0, \end{aligned} \tag{C.5}$$

which is valid for $n + 1 \leq p \leq 2n$. Therefore we have $a_0 \eta_p + \sum_{k=1}^{n} a_k \eta_{p-k} = 0$, which can be expressed as an $n \times n$ matrix equation:

$$\begin{pmatrix} \eta_n & \eta_{n-1} & \cdots & \eta_1 \\ \eta_{n+1} & \eta_n & \cdots & \vdots \\ \vdots & \vdots & \ddots & \vdots \\ \eta_{2n-1} & \eta_{2n-2} & \cdots & \eta_n \end{pmatrix} \begin{pmatrix} a_1 \\ a_2 \\ \vdots \\ a_n \end{pmatrix} = - \begin{pmatrix} \eta_{n+1} \\ \eta_{n+2} \\ \vdots \\ \eta_{2n} \end{pmatrix}. \tag{C.6}$$

Solving this matrix equation gives the coefficients $a_k$ of the characteristic polynomial $\phi(\lambda)$, and then its roots $\lambda_l$ can be calculated. Substituting $\lambda_l$ back into Eq. (C.2) and solving it to get the coefficients $\alpha_l$, and the desired expression (C.1) is obtained.

To this end, we note that the Prony method has an intimate relation to the linear prediction technique used to extrapolate the Green's function to longer times [75]. Both techniques assumes that the underlying function can be approximately decomposed as in Eq.(C.1). The difference is that the Prony algorithm explicitly finds the coefficients of the decomposition. While in linear prediction, one is interested in the recursion relation

$$\tilde{\eta}_m = -\sum_{i=1}^{p} b_i \eta_{m-i},\tag{C.7}$$

between the history and the future, therefore the goal is to determine the values of $b_i$ by the least square method. As discussed in Ref. [75], the linear prediction works well for Green's function because the internal structure of Green's function has the form in Eq.(C.1). In fact, after some transform one could derive Eq.(C.7) from Eq.(C.1). Thus in principle, in linear prediction one could also use the Prony algorithm to find the explicit decomposition in Eq.(C.1) first, and then use it to predict future results instead of finding the recursion relation in Eq.(C.7).

# D   Memory truncation versus the zipup algorithm

The computational costs of both the partial IF and TTI IF approaches are dependent on $t$, partially because that the ranks of the involved Grassmann tensors scale linearly as $N$. In the bosonic case, a memory truncation scheme is adopted in QuaPI and TEMPO to reduce the scaling of the computational cost [33, 61]. The basic idea of the memory truncation scheme is as follows: the hybridization function $\Delta^{\zeta\zeta'}(t, t')$ vanishes as $|t - t'| \to \infty$, accordingly the discretized hybridization function $\Delta_{j,k}^{\zeta\zeta'}$ also vanishes as $|j - k| \to \infty$, therefore we may truncate the discretized hybridization function when $|j - k|$ is large enough. In numerical calculations, one may set $\Delta_{j,k}^{\zeta\zeta'} = 0$ when $|j - k| > \Delta k_{\max}$, with $\Delta k_{\max}$ a positive integer. As a result, one only needs to keep $\Delta k_{\max}$ time steps of the ADT, and the ADT can be evolved to later time iteratively via tensor multiplications and integrating out previous time steps further apart than $\Delta k_{\max}$. Concretely, we illustrate the first step of the memory truncation scheme with $\Delta k_{\max} = 3$ in Fig. 6(a). In the beginning, we have the initial ADT with four time steps from $t_0$ to $t_3$, represented by light purple circles. Then we need to apply a tensor (here "apply" means the element-wise product in Appendix. B), which represents a partial IF from $t_1$ to $t_4$ ($t_0$ is absent in the partial-IF because its distance to $t_4$ is larger than $\Delta k_{\max}$). In the meantime the bare impurity dynamics is involved from $t_3$ and $t_4$. After the tensor multiplication, the $t_0$ step is integrated out which yields a new ADT which only consists of time steps from $t_1$ to $t_4$.

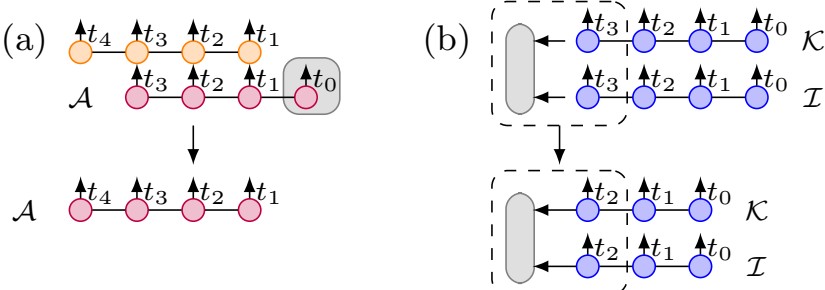

Figure 6: Schematic illustration of (a) the first step of the iterative construction of the ADT using the memory truncation scheme and (b) the zipup algorithm for Toulouse model. The gray rectangular in (b) represents an environment tensor obtained by iteratively integrating out the GVs in each time step.

It can be seen that in the memory truncation scheme, the ADT is explicitly built. Assuming that the bond dimension of the ADT is $\chi_{\mathcal{A}}$, then the computational cost of building the ADT roughly scales as $O(N\Delta k_{\max}\chi_{\mathcal{A}}^3)$, in comparison with the computational cost $O(N^2\chi^3)$ of the partial IF approach for building the MPS-IF.

In GTEMPO, we do not construct the ADT explicitly because $\chi_{\mathcal{A}}$ could be very large, even though the bond dimension $\chi$ of each MPS-IF may be small. Instead we use a zipup algorithm which calculates the ADT only on the fly when calculating multi-time correlations, as illustrated in Fig. 6(b) for the Toulouse model. We denote the bond dimension of $\mathcal{K}$ as $\chi_{\mathcal{K}}$. The ADT can be obtained by multiplying $\mathcal{K}$ and $\mathcal{I}$. In general we found that the bond dimension of the ADT can be hardly reduced by MPS compression (or one may result in significant loss of accuracy [42]). Therefore the bond dimension of ADT would simply be $\chi_{\mathcal{A}} = \chi_{\mathcal{K}}\chi$ for Toulouse model (and $\chi_{\mathcal{A}} = \chi_{\mathcal{K}}\chi^2$ for the single-impurity Anderson model). Even for the simple case of the SIAM, we can see the huge gain of the zipup algorithm compared to explicitly building the ADT, since the computational cost of the latter scales at least as $O(\chi^6)$ (more details of the zipup algorithm can be found in Refs. [41, 42]). However, one disadvantage of the zipup algorithm is that it is essentially incompatible with the memory truncation scheme, which can also be seen from Fig. 6(b): if one integrates out the time step $t_0$, the two MPSs for $\mathcal{K}$ and $\mathcal{I}$ will be joint together, then one would result in a single MPS representation of the ADT, which becomes equivalent to the memory truncation scheme.

Now we can have a thorough discussion of the memory truncation scheme and the zipup algorithm in terms of their computational costs. In the fermionic case, one generally faces a large number of "flavors", for example, even in the simple case of the SIAM, one has two flavors, for spin up and spin down respectively. In such situation, the zipup algorithm is the method of choice, even though without the memory truncation the computational cost of constructing $\mathcal{I}$ scales with $t^2$ (the partial IF method) or $t$ (the TTI IF method). In comparison, in the bosonic case, one usually considers a single flavor. For example, for the spin-boson model, one has $\chi_{\mathcal{K}} = 2$ and thus $\chi_{\mathcal{A}} = 2\chi$ at most. In this case the memory truncation scheme would be as efficient as as the TTI IF approach as its cost only scales linearly with $t$. Here we also point out that even in such ideal situation, the memory truncation scheme still has several drawbacks: (i) the hyperparameter $\Delta k_{\max}$ requires deep knowledge of the model, which is generally hard to be determined before hand; (ii) it can not be used in the imaginary-axis calculations as the hybridization function does not decay except for zero temperature.

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
