# Peer review of "Efficient construction of the Feynman-Vernon influence functional as matrix product states"

_SciPost Physics, doi:SciPost Phys. Core 7, 063 (2024)_

## Round 2 · Referee Report · Anonymous (Referee 1) · 2024-6-12

Strengths
Weaknesses
just based on this paper itself
Report
thank you for forwarding the manuscript by Guo et al. on the
Efficient construction of the Feynman-Vernon influence
functional as matrix product states (2402.14350v2).
The paper aims to study the real-time dynamics of quantum
impurity problems. The bath is cast into an effective
description, allowing the authors to treat
the impurity in an effective environment.
Quantum impurity models represent a long-standing important
class of problems in condensed matter. Hence it is interesting
to see new developments. The authors, however, already have
three other papers along this line already within the last half
a year, all with very similar titles:
[41] R. Chen, X. Xu, and C. Guo,
Grassmann time-evolving matrix product operators for
quantum impurity models, PRB (2024).
[42] R. Chen, X. Xu, and C. Guo,
Grassmann time-evolving matrix product operators for equilibrium
quantum impurity problems, New J. Phys. (2024).
[43] R. Chen, X. Xu, and C. Guo,
Real-time impurity solver using grassmann time-evolving matrix
product operators, arXiv:2401.04880 (2024).
This raises the question of whether the present paper
represents a rather incremental or even duplicate step.
Having several similar near-simultaneous papers and
referencing these, makes the present paper rather unreadable
by itself. Figures like Fig. 1 are not understandable
the way they are currently presented.
The paper is aware of other recent works on MPS representation
of the Feynman-Vernon influence functional (Refs. 47-51).
However, the precise overlaps of and differences with these
other approaches are not discussed much at all.
A few lines in the introduction would be helpful.
It appears there is a distinction based on a many-body
treatment vs. Gaussian-like states which effectively
can work in a single-particle picture. This needs to
be explained in more detail.
Hence while the present paper appears to have merit,
I strongly recommend that the authors address in detail
the points raised below in the paper itself prior
to me deciding on a recommendation for its publication.
More detailed remarks:
Overall, I find it difficult to follow the arguments.
In Eq. (5): why do the partial IFs commute with each other?
After all, k is summed over, and the a's anticommute.
Where does Eq. (7) come from?
Fig. 1 is not understandable without having to look up
other references of the author, like Refs 41-43.
And even then I am not sure what is shown. Naively,
MPS cannot be muliplied unless one calculates an overlap.
But multiplying them to get another MPS? That usually requires MPOs.
Do the authors have tensor product in mind?
Also, how does this relate to a Keldysh contour?
The bath of an impurity is described by a continuum of states.
Hence from a numerical perspective, I'd expect this needs to
be discretized, only then resulting in discrete Grassmann variables.
But this also comes with a discretization error that appears
discussed nowhere. This seems hidden behind the reference
to the `QuaPI' method (besides, should this rather be `QuAPI'?)
For ED, the paper mentions uniform discretized of the
bath into 8000 intervals. How does this compare to GTEMPO?
If there is a coarse-graining of the bath in GTEMPO,
does this coarse-graining have to be uniform over the
spectral range? Or may one choose a logarthmic grid
as common for the numerical renormalization group (NRG)
for the benefit of reaching much lower energies /
longer time scales?
Note that there are also mixed log-linear discretization
schemes used in the literature for non-equilibrium
transport through correlated impurities.
The paper writes: `With the more efficient TTI IF method
to construct the MPS-IF, we can easily reach longer
evolution time' [up to t=8.4/Gamma]. For impurity models
which can have an exponential dynamically generated
low-energy scale (the Kondo temperature)
this is still a very modest time range.
What is the limiting case in time here?
The paper talks about steady-state currents, but Fig. 4
still mostly shows transient behavior.
If t=8.4/Gamma `can be easily reached', why not go to
much longer times to reach a steady state?
It would be helpful to quantitatively compare to
steady-state currents for interacting quantum impurity models
as found already in the literature, e.g., PRL 101, 140601 (2008),
or PRL 121, 137702 (2018).
Also why not show the full transient in Fig. 4 from t=[0,8]/Gamma
rather than only the second half [4,8]?
Minor details:
The paper writes: `we note that our method can be directly
applied to general QIMs as long as the Feynman-Vernon IF
applies'. Please be more specific: When does it not apply?
Does this require a non-interacting bath?
After Eq. (4), the `bath spectrum density',
by its name, appears a bath property, and therefore
should be independent of impurity parameters such as Vk
which already refers to hybridization.
It appears to me that the Prony algorithm is closely
related to `linear prediction'. The authors may
briefly comment on the relation between the two
in their Appendix A.
Requested changes
(see report above)
Recommendation
Ask for major revision

Author: Ruofan Chen on 2024-06-24 [id 4580]
(in reply to Report 1 on 2024-06-12)Reply to Referee
The referee writes:
Our response: We thank the Referee for refereeing this work and for his/her comments. The previous works focus on impurity solvers using our newly propose GTEMPO framework. Concretely, Ref.[41] solves non-equilibrium impurity problem on the real-time axis. Ref.[42] solves the equilibrium impurity problem on the imaginary-time axis. Ref.[43] extends the method in Ref.[41] to solve equilibrium impurity problem on the real-time axis with a quench. In those works, a central numerical calculation is to build the influence functional as a GMPS, which is done using the partial-IF method. The number of GMPS multiplications required in the partial-IF algorithm scales as O(N) for N discrete time steps. The current work does not aim for any new impurity solver. Instead, it aims to replace the partial-IF algorithm with a more efficient algorithm, referred to as the TTI-IF method, to build the influence function as a GMPS. Therefore, one could use the TTI-IF method in all the previous works to significantly improve the numerical efficiency. Compared to the partial-IF algorithm, the TTI-IF method respects the time-translational invariance of the influence functional, and the number of GMPS multiplications required in the new method does not scale with N. As we understand from the referee’s comment, the difficulty of understanding Fig.1 is mainly due to that the GMPS multiplication is not well explained in this work. The multiplication of two GMPSs result in a new GMPS, which is different from usual MPO or MPS arithmetic. This operation is introduce in Ref.[41], to make the current work more self-contained, we have added a short explanation of this operation in Appendix. A. Roughly speaking, the GMPS multiplication originates from the Grassmann tensor multiplication, which is the only operation required to build the MPS-IF. We also mention that the bosonic counterpart of the Grassmann tensor multiplication is the element-wise product of two normal tensors. In principle, one could also convert the MPSs into MPOs by “copying” its physical indices, and the GMPS multiplication will be converted into MPO multiplications which is a standard operation. But we prefer to directly implement the GMPS multiplication for efficiency without the intermediate conversion step (the element-wise operation is more natural for manipulating the path integral).
The referee writes:
Our response: We thank the referee for this comment. We have added in the introductory section a brief discussion about the difference between our GTEMPO and the tensor network IF method, which is attached as follows:
Briefly speaking, in the tensor network IF method, the Grassmann path integral is converted into a fermionic operation expression, which thus avoids to directly deal with Grassmann variable numerically. However, the addition transformation would make it less convenient for numerical implementation. Moreover, as shown in Ref.[41], for transport problem the computational cost of the tensor network IF method scales exponentially against the number of baths, while the cost of GTEMPO is essentially independent of the number of baths.
The referee writes:
Our response: We thank the referee for this comment. This is because that any Grassmann expression with an even number of Grassmann variables commute with each other, which is added in the end of Page 4 in the revised manuscript. Thinking of Eq.(7) requires some intelligence and experience. However, it is easy to verify that it is correct: One can simply do the multiplication of the right hand side of Eq.(7) and will find that the result is exactly Eq.(6).
The referee writes:
Our response: We thank the referee for this comment. As in our answer to comment (1), we think that the gap in understanding Fig.1 is to understand the definition of GMPS multiplication, which results in a new GMPS. We have thus added in Appendix.A the definition of GMPS multiplication, which is the only operation needed to build the MPS-IF.
The referee writes:
Our response: We thank the referee for this comment. We would like to clarify some misunderstanding of the referee about the GTEMPO method here. First of all, the major difference of GTEMPO, compared to conventional MPS methods, is that the bath degrees of freedom is analytically integrated out in GTEMPO using the Feynmann-Vernon influence functional, and one is only left with the impurity degrees of freedom at different times, which is represented as a GMPS. Thus there is no bath discretization error in GTEMPO, the major sources of error are the time discretization and the MPS bond truncation error. Second, the QuAPI method does not handle the bath discretization neither. It merely discretize the continuous hybridization function \Delta into a discrete hybridization matrix used in the exponent of Eq.(4). We also thanks the referee for pointing out the mistake of abbreviation QuaPI to QuAPI, we have corrected it in the article.
The referee writes:
Our response: We thank the referee for this comment. The computational cost of the TTI-IF method scales as O(N\chi^4) as pointed in the manuscript. For t=8.4/\Gamma with \delta t=0.014/\Gamma, we have N=600. And since there are 8 Grassmann variables per time step, the size of the GMPS is about 4800 already in our numerical simulations. It is easily to do better than the partial-IF but the cost will still be very significant if we further enlarge t. Therefore we stop at t=8.4/\Gamma. In the revised manuscript we have changed the sentence “we can easily reach longer evolution time” into “we can reach longer evolution time” to weaken the statement.
The referee writes:
Our response: We thank the referee for this comment. The reason that we do not go to even larger times is answered in our reply to comment (6): there is a linear scaling of the cost with t, and even with current choice of t the size of GMPS is close to 5000, thus it is hard to consider a much larger t and we decide to stop at this value. We also note the numerical simulation in Fig.4 does not aim particularly for the steady state. In fact, one would still need algorithmic development if one is interested in the steady state only, since the time to reach steady state may be very long with the current method, which can be seen from Fig.4(a,b,c), especially for small chemical potential bias. For example, one could use infinite MPS techniques which directly targets at the steady state and completely neglects the transient dynamics (our later work, arXiv: 2403.16700 extends the technique of the current work to directly aim for the steady state). Similarly, benchmarking the steady state currents with “PRL 101, 140601 (2008), or PRL 121, 137702 (2018)” is likely to be very challenging for the current method, which is left to investigate in future developments. To stress the scope of our numerical simulation of the transport problem, we have added the sentence “Our goal in this study is mainly to verify whether the existing finite-time calculations have reached steady state or not, by extending the evolution time with our new method” in the revised manuscript. From Fig.4, we can clearly see that the answer to this question is no for small V < 0.71 and is yes otherwise.
The referee writes:
Our response: We thank the referee for this comment. We start from t=4.2/\Gamma, as the existing finite-time calculations stop here, and our goal is to verify whether these existing finite-time calculations has reached steady state or not. We have added this sentence in the revised manuscript, which is attached as follows:
The referee writes:
Our response: We thank the referee for this comment. We have added an explanation after this sentence, which is attached in the following:
The referee writes:
Our response: We thank the referee for this comment. The name of this function differs from literature to literature. We agree that the name “bath spectrum density” is misleading as it not only contrains the bath information. We have replaced it by “coupling strength function”, which has been used in [Erpenbeck et. al., Phys. Rev. Lett. 130, 186301 (2023)].
The referee writes:
Our response: We thank the referee for this comment. These two algorithms are indeed intimately connected: both of them learn the function in Eq.(B4). The difference is that in the Prony algorithm Eq.(B4) is explicitly used (the parameters in this equation is directly used to construct the MPO afterwards), while in linear prediction, one often does not need the explicit form of Eq.(B4), but only the recursion relation between the history and future, which is implicitly determined by Eq.(B4). Of course, in linear prediction, one can also obtain the explicit form of Eq.(B4) first and then use it to predict the future, instead of predicting the future based on a recursion from the history data. We have added this discussion in the end of Appendix.B in the revised manuscript.

---

## Round 3 · Referee Report · Anonymous (Referee 1) · 2024-7-16

Report

I thank the referees for addressing most of my concerns. The addition of Appendix A was very helpful, indeed (see more detailed comments on this further below).

Indeed I missed that the approach eventually concerns the impurity only, with the bath fully integrated out. As an aside then, in Eq. (3) the a's and \tilde{a}'s are only introduced as `Grassmann trajectories'. It shall also be emphasized then that these refer to the impurity only, even though indirectly implied by the notation and context.

So the Grassmann tensors are MPS/MPOs of length N=t/dt. Please specify max(N) in the paper as used for the paper, even though one may imply it from the data. Similarly, please include the response to my question on reachable time scales in the paper itself, e.g., N=600 leading to L=8N Grassmann variables. Is L the actual length of the MPS then?

With respect to Fig. 1 then, MPS multiplication of GTs considers all i=1,..,N `sites' = time steps. This appears to imply that to reach some final time t, the whole time windows [0,t] needs to be addressed simultaneously. This appears in stark contrast to wave function based approaches, where one iteratively proceeds from time t -> t+dt. It would be useful to briefly address this in the paper.

The parameters for truncation like \chi are set by hand, and convergence is checked by comparing results. However, to have a sensible estimate of the numerical cost, the following two questions need to be addressed, assuming sufficiently large \chi:

1) how does the entanglement entropy of the target MPS scale with N=t/dt for a fixed time window [0,t]? (i.e., making dt smaller)

2) how does the entanglement entropy of the target MPS scale with increasing t in [0,t] for fixed dt = t/N?

The argument of linear scaling of the cost with t relies crucially on the assumption that the entanglement entropy in (1) and (2) saturates, so that the same fixed \chi may be used. However, is this justified? The answer will depend sensitively on the answer to (1) and (2) above.

Detailed remarks on App. A: Multiplication of Grassmann tensors (GTs)

In (A3), it appears the upper indices with \xi_n^{i_n} are actual powers, not contour indices as in the main text. Please make sure this is understood unambiguously.

Typo in line after (A5): x_1 -> \xi_1

The last paragraph in App. A is rather obscure. It may be replaced by a discussion along the following:

It appears to me that (A7) can be compactly formulated as a rank-3 fusion tensor, say X, that can be simply contracted onto a pair of matching Grassmann variables in the product of two MPS/MPOs.

Furthermore, when having an MPS representation of a GT, this likely can make use of a Z_2 parity structure to count number of particles (GVs) mod2. Then all A-tensors along the MPS are parity preserving, such that sign-strings can be efficiently incorporated in terms of parity gates on the MPS bond next to the respective site.

If the above is correct, then it appears to me that the multiplication of the GTs represented as MPS should have a simple transparent pictorial tensor network representation that includes fusion tensors X and parity gates on MPS bonds.

Corollary to the above: did the authors exploit Z_2 symmetry in their MPS/MPO simulations, or did they use full tensors throughout? This is important for interpreting numerical costs. Please clarify in the paper itself.

Requested changes

see report above.

Recommendation

Ask for minor revision

  • validity: good
  • significance: good
  • originality: good
  • clarity: ok
  • formatting: good
  • grammar: good

Author:  Ruofan Chen  on 2024-08-09  [id 4682]

(in reply to Report 1 on 2024-07-16)

We thank the referee for the refereeing and comments. Our response to the comments are shown below.

The referee writes:

Indeed I missed that the approach eventually concerns the impurity only, with the bath fully integrated out. As an aside then, in Eq. (3) the a's and \tilde{a}'s are only introduced as `Grassmann trajectories'. It shall also be emphasized then that these refer to the impurity only, even though indirectly implied by the notation and context.

Our response: We thank the referee for the suggestion, and we have added a sentence after Eq. (3) that

It should be noted that this path integral formalism only contains the impurity GVs in the temporal domain.

The referee writes:

So the Grassmann tensors are MPS/MPOs of length N=t/dt. Please specify max(N) in the paper as used for the paper, even though one may imply it from the data. Similarly, please include the response to my question on reachable time scales in the paper itself, e.g., N=600 leading to L=8N Grassmann variables. Is L the actual length of the MPS then?

Our response: We thank the referee for the suggestion. Since in a time step there are 8 GVs (two spins, two branches, and the conjugate), the actual length of the MPS is indeed L=8N for the SIAM. We have added two sentences, one after Eq. (4)

Since there are 8 GVs within each time step, the total number of GVs is 8N

and one in the first paragraph of Sec.II B

In our implementation we represent each GV as one site, therefore the MPS representations of K and I all have 8N sites for the SIAM.

to specify this point. The max(N) used in all our simulations is N=1200 corresponding to Dt=120 with delta t=0.05, which has also been specified after Eq.(18).

The referee writes:

With respect to Fig. 1 then, MPS multiplication of GTs considers all i=1,..,N `sites' = time steps. This appears to imply that to reach some final time t, the whole time windows [0,t] needs to be addressed simultaneously. This appears in stark contrast to wave function based approaches, where one iteratively proceeds from time t -> t+dt. It would be useful to briefly address this in the paper.

Our response: We thank the referee for the suggestion. We have discussed this point after Eq. (7), which is attached in the following:

Here we can also see another stark difference between the GTEMPO and the wave-function based MPS methods that in GTEMPO the whole evolution time widow [0,t] is addressed simultaneously, while in the latter the evolution proceeds from time t to t + dt iteratively. Furthermore, in GTEMPO we first build the MPS-IF for the whole time interval [0, t], which is then used for calculating any multi-time impurity correlations within this time interval.

The referee writes:

The parameters for truncation like \chi are set by hand, and convergence is checked by comparing results. However, to have a sensible estimate of the numerical cost, the following two questions need to be addressed, assuming sufficiently large \chi:

1) how does the entanglement entropy of the target MPS scale with N=t/dt for a fixed time window [0,t]? (i.e., making dt smaller)

2) how does the entanglement entropy of the target MPS scale with increasing t in [0,t] for fixed dt = t/N?

The argument of linear scaling of the cost with t relies crucially on the assumption that the entanglement entropy in (1) and (2) saturates, so that the same fixed \chi may be used. However, is this justified? The answer will depend sensitively on the answer to (1) and (2) above.

Our response: We thank the referee for the comment. We have added discussions in the last paragraph of Sec.IIIA to justify the effectiveness of the MPS representation of the IF. As suggested by the referee, we have also added Fig.4 in the revised manuscript, which shows the scaling of the operator space entanglement entropy (OSEE, which is the bipartition entanglement entropy of density operators), as a function of t for two different values of dt. We can see that the OSEE indeed approximately saturates for large t, which we think can well answer question 2) of the referee. For question 1), we have considered two different values of dt in the two panels of Fig.4, and shown that the accuracy increases for smaller dt. However, we note that there is no known theoretical guarantee that the OSEE would saturate against smaller dt. To balance numerical stability and accuracy, in both GTEMPO and the wave-function based MPS methods, one often chooses a reasonable value of dt (not too small and not too large) in practice.

The referee writes:

In (A3), it appears the upper indices with \xi_n^{i_n} are actual powers, not contour indices as in the main text. Please make sure this is understood unambiguously.

Our response: We thank the referee for the suggestion. We have added a sentence after (A4) [now (B4)] to remove the unambiguity, which is attached as follows:

where the upper indices i_k,j_k are actual powers.

The referee writes:

Typo in line after (A5): x_1 -> \xi_1

Our response: We thank the referee for pointing out the typo, it has been corrected.

The referee writes:

The last paragraph in App. A is rather obscure. It may be replaced by a discussion along the following:

It appears to me that (A7) can be compactly formulated as a rank-3 fusion tensor, say X, that can be simply contracted onto a pair of matching Grassmann variables in the product of two MPS/MPOs.

Furthermore, when having an MPS representation of a GT, this likely can make use of a Z_2 parity structure to count number of particles (GVs) mod2. Then all A-tensors along the MPS are parity preserving, such that sign-strings can be efficiently incorporated in terms of parity gates on the MPS bond next to the respective site.

If the above is correct, then it appears to me that the multiplication of the GTs represented as MPS should have a simple transparent pictorial tensor network representation that includes fusion tensors X and parity gates on MPS bonds.

Corollary to the above: did the authors exploit Z_2 symmetry in their MPS/MPO simulations, or did they use full tensors throughout? This is important for interpreting numerical costs. Please clarify in the paper itself.

Our response: We thank the referee for the comment. We reply to the comments of the referee point by point in the following:

It appears to me that (A7) can be compactly formulated as a rank-3 fusion tensor, say X, that can be simply contracted onto a pair of matching Grassmann variables in the product of two MPS/MPOs.

(A7) [now (B7)] can be implemented as rank-3 tensor, but is not the usual fusion tensor. The two input spaces are of size 2, and the output space also has size 2. In comparison a usual fusion tensor should have output space size 4. Nevertheless, if we explicitly implement this operation as the contraction with a rank-3 tensor, the code could look more elegant.

Furthermore, when having an MPS representation of a GT, this likely can make use of a Z_2 parity structure to count number of particles (GVs) mod2. Then all A-tensors along the MPS are parity preserving, such that sign-strings can be efficiently incorporated in terms of parity gates on the MPS bond next to the respective site.

Yes, this is what we do in our actual implementation, which has been thoroughly discussed in Ref.[41].

If the above is correct, then it appears to me that the multiplication of the GTs represented as MPS should have a simple transparent pictorial tensor network representation that includes fusion tensors X and parity gates on MPS bonds

Yes, this can be done. But in our current implementation we have not used the rank-3 “fusion tensor” yet.

Corollary to the above: did the authors exploit Z_2 symmetry in their MPS/MPO simulations, or did they use full tensors throughout? This is important for interpreting numerical costs. Please clarify in the paper itself.

Yes, we have used the Z_2 symmetric tensor in our GMPS simulations, which can reduce the global sign changes into local ones.

Overall, we have stressed that in our actual implementation we used the Z_2 symmetric MPS. And we have also pointed to Ref.41 for interested readers.

---

## Round 3 · Referee Report · Anonymous (Referee 2) · 2024-7-22

Strengths

1- Succinct and well written 2- Advances the influence functional approach to calculating the dynamics of open quantum systems by means of well-known tools from tensor networks

Weaknesses

1- The overall impact of the work is somewhat obscure 2- Small grammatical mistakes throughout the paper

Report

The authors make a case for how to simplify, both computationally and conceptually, the influence functional (IF) approach to quantum impurity problems, and in doing so suggests a unified language for constructing the influence functional in problems involving quadratic bosonic and fermionic environments. This is done by the introduction of the "time-translationally invariant IF" (TTI) method, in which the IF is built from matrix product states (MPS) in which the site tensors are the same, modulo boundaries. The authors show that the TTI has lower scaling runtimes compared to the partial IF as a function of total propagation time, and tests the method by calculating the current through single Anderson impurity coupled to two leads.

The authors' approach leverages well-known ideas of the tensor network community. Here, long range interactions (in time) are approximated using sums of exponentials which allow for simple matrix product operator (MPO) representations, and exponentials of matrix product operators are computed using W(I)/W(II) approaches. In this paper MPO-MPS contractions are replaced in favor of MPS-MPS multiplications.

The overal claim that the authors make is that their TTI method is computationally superior to their partial IF method. Overall, this seems like a somewhat weak result from several perspectives: 1- While this claim on scaling may true, it relies on the number of time steps $N$ being larger than the considered bond dimension $\chi$. This is due to the overall scaling of the partial IF method being $O(N^2 \chi^3)$ versus the overall scaling of the TTI method being $O(m N \chi^4)$, where $m$ is the number of times the MPS is "squared". In the same spirit as QUAPI or TEMPO, by making use of a truncation of $\Delta^{\zeta \zeta'}_{j,k}$ such that this is set to zero for $|j-k| > N_{trunc}$, would it not be possible to reduce the scaling of the partial IF method to $O(N_{trunc} N \chi^3)$? How much would this affect the accuracy of the dynamics? If the overall effect is not too large, then would the central claim of the paper need to be weakened? 2- The authors mention that there are other recent papers that construct the IF for fermionic impurity problems though without using the Grassmann MPS formalism. At the very least it would be useful to make some comparison of efficiency against those works to at least contextualise the authors' claim. 3- IF is not the only method for solving quantum impurity problems. How does the TTI method fare against a more straightforward approach, say, with using MPS to represent wavefunctions of a discretised bath?

There is a way in which this work seems incremental, even from the term "time-translationally invariant IF". This term suggests that the IF is considered as a truly time-translationally invariant object, and so one would expect it to be infinite in extent so that it can be represented as a uniform MPS. The benefits to the scaling of the method would become immediately apparent, as there would be no dependence on $N$. This is the approach taken in a paper cited by the authors, ref. 71 by Link et al. It is rather perplexing to construct this TTI method without taking full advantage of the stated time-translational invariance. That the authors have in the subsequent month or two posted a paper fully using this time-translationally invariant property only enhances this sense of incrementality.

In summary, this paper has merits that may be of interest to those working in open quantum systems. As written, however, the overall impact of this work is weakened as it only narrowly focuses on an influence functional approach, and for that matter, only on the authors' previous method. While this is not a sin per se, it would still behoove the authors to enhance the clarity of their paper by giving more details about their calculations.

Detailed remarks: 1- It would be useful to include a plot of how the required bond dimension to converge the calculation increases as a function of time. 2- What is the temperature used for the Toulouse model calculations? It should be the case that the retarded Green's function is temperature independent, but the IF and the difficult in constructing it should depend strongly on the temperature. 3- How practical is the TTI method in absolute terms, i.e. what is the runtime? If the Toulouse model example with $\chi = 50$ is any indication, constructing the IF for the SIAM with $\chi = 160$ will require 100x more time. 4- The paper mentions the Fishman-White algorithm, which constructs the Gaussian state of the IF by applying local gates to the vacuum state (a method to construct the partial IF). In this algorithm, the number of tensor network operations also scales roughly linearly with time. Does this imply that there is a way to construct the partial IF with similar efficiency scaling as the TTI method, even without leveraging the time translational invariance of the hybridization terms? 5- For all the simulations, please specify whether you are computing the IF up to time $t_{max}$ and extracting observable values for all $t <= t_{max}$, or whether you are computing observable values at time t using the IF defined to time $t$. 6- Regarding Figure 4, can the authors comment on why the earlier times (t=4) are less accurate than longer times when compared to exact diagonalization results? 7- Fig 3b at time $Dt/2 < 5$ shows that improving the prony fitting error does not necessarily lead to similar order of magnitude improvements in the dynamics. Can the authors comment on why this is the case for the Lorentzian coupling strength function as opposed to the semi-circular one? Is the latter easier due to the finite support over $\omega$? Does the error in the retarded Green's function reflect the deviation of the Prony fit?

Minor remarks: 1- In Section II.B, is the statement "Based on the ADT, one can easily calculate any multi-time correlations of the impurity" accurate? Following the definition of the ADT from QUAPI and TEMPO, the ADT should already factor in system propagations as well. That is, the ADT is entire integrand of eq (3), as opposed to the IF which is only the $I_{\sigma}$ part of the integrand. Once the system propagation tensors $K$ are included, is it possible to freely calculate any multi-time correlations? 2- Related to the previous point, the symbol $K$ in eq (3) is not defined in any of the text surrounding the equation. 3- It would be useful to include an additional appendix with the explicit expressions of the hybridization terms, as this would make the paper much more self-contained. 4- Is there a technical reason why the authors only consider the IF up to first order error in the time discretization? 5- In the sentence below eq (9), it seems to imply that the Prony algorithm finds the optimal parameters for the exponential fit. Is this actually true?

Requested changes

See report

Recommendation

Ask for major revision

  • validity: good
  • significance: ok
  • originality: ok
  • clarity: good
  • formatting: excellent
  • grammar: excellent

Author:  Ruofan Chen  on 2024-08-09  [id 4683]

(in reply to Report 3 on 2024-07-22)

We thank the referee for the refereeing and comments. Our response to the comments are shown below.

The referee writes:

While this claim on scaling may true, it relies on the number of time steps N being larger than the considered bond dimension χ. This is due to the overall scaling of the partial IF method being O(N^2χ^3) versus the overall scaling of the TTI method being O(mNχ4^4), where m is the number of times the MPS is "squared". In the same spirit as QUAPI or TEMPO, by making use of a truncation of Δζζ′j,k such that this is set to zero for |j−k|>Ntrunc, would it not be possible to reduce the scaling of the partial IF method to O(NtruncNχ^3)? How much would this affect the accuracy of the dynamics? If the overall effect is not too large, then would the central claim of the paper need to be weakened?

Our response: We thank the referee for the comment. The memory truncation scheme is a key technique to reduce the computational scaling in QUAPI and TEMPO method. However, such a scheme is based on the manipulation of ADT which is obtained by multiplication of K and I. The bond dimension of the ADT, \chi_{A}, is then roughly the product of those of K, denoted as \chi_{K}, and I, denoted as \chi in the paper, which can easily be very large to overwhelm any computational gain. In the spin-boson model there is no problem as there is only a single flavor and \chi_{K}=2, but the problem will be serious in the fermionic case. Therefore in GTEMPO, we never compute the ADT explicitly but to integrate the ADT only on the fly with separate K and I using the zipup algorithm. In the meantime, the zipup algorithm is incompatible with the memory truncation scheme, therefore we opt to use the zipup algorithm. We have added appendix D to thoroughly discuss the memory truncation scheme and the zipup algorithm.

The referee writes:

The authors mention that there are other recent papers that construct the IF for fermionic impurity problems though without using the Grassmann MPS formalism. At the very least it would be useful to make some comparison of efficiency against those works to at least contextualise the authors' claim.

Our response: The major performance advantage of GTEMPO over the tensor network IF methods is that the computational cost of GTEMPO is essentially independent on the number of baths, as GTEMPO is purely dependent on the Feynman-Vernon IF in which the baths are integrated out, while the cost of the tensor network IF method reported in Ref.48 scales exponentially with number of baths. We have added a sentence in the end of the first paragraph of Sec.III B to state this difference when studying the transport problem.

The referee writes:

IF is not the only method for solving quantum impurity problems. How does the TTI method fare against a more straightforward approach, say, with using MPS to represent wavefunctions of a discretised bath?

Our response: The IF based method has already integrated out the bath degrees of freedom, thus it is free of bath discretization error compared to the wavefunction based method. We have discussed this issue in the end of third paragraph of the introduction. And the TTI IF method described proposed in this work aims to improve the construction efficiency of the MPS-IF.

The referee writes:

There is a way in which this work seems incremental, even from the term "time-translationally invariant IF". This term suggests that the IF is considered as a truly time-translationally invariant object, and so one would expect it to be infinite in extent so that it can be represented as a uniform MPS. The benefits to the scaling of the method would become immediately apparent, as there would be no dependence on N. This is the approach taken in a paper cited by the authors, ref. 71 by Link et al. It is rather perplexing to construct this TTI method without taking full advantage of the stated time-translational invariance. That the authors have in the subsequent month or two posted a paper fully using this time-translationally invariant property only enhances this sense of incrementality.

Our response: We thank the referee for noticing our subsequent work [Guo and Chen, Phys. Rev. B 110, 045106 (2024)], where the infinite GTEMPO (iGTEMPO) method is represented. The iGTEMPO method employs the infinite MPS technique which takes the full advantage of time-translational invariance and its cost is essentially independent on t. In the meantime, the work by Link et al. is also a very nice work which made use of the time translational invariance but in a completely different way (in the language of our work, this latter approach found a way to explore the TTI property based on the partial IF approach). It would be interesting to make a comparison between these two different approaches to build the time translationally invariant MPS-IF in the future.

In the meantime, we donot think this work is only incremental. The contribution of this work at least includes: (1) it lays down the central techniques used for building the full time translationally invariant MPS-IF used in our later work; (2) it compares in detail the performances of the partial IF and the TTI IF methods; and (3) most importantly, the transient dynamics is an important subject in its own right, for example it is a central ingredient in non-equilibrium DMFT, and the transient dynamics will be lost if infinite MPS is used. Therefore we think that speeding up the construction of the MPS-IF for the transient real-time dynamics itself merits an individual work.

The referee writes:

It would be useful to include a plot of how the required bond dimension to converge the calculation increases as a function of time.

Our response: We thank the referee for the comment. The first referee raised a similar comment. In this work our strategy to compress the MPS is to use a fixed bond dimension \chi. Therefore a more meaningful quantity is the operator space entanglement entropy (OSEE) which characterizes the growth of the bipartition entanglement of the MPS-IF (which is similar to a density operator). We have added Fig.4 in the revised manuscript to show the growth of OSEE against the total evolution time t, from which we can see that indeed the OSEE approximately saturates for large t.

The referee writes:

What is the temperature used for the Toulouse model calculations? It should be the case that the retarded Green's function is temperature independent, but the IF and the difficult in constructing it should depend strongly on the temperature.

Our response: We thank the referee for the comment. The inverse beta used is 20 which is now stated after Eq.(19). In principle the temperature would affect the memory length of the hybridization function, and thus affect the bond dimension of MPS-IF.

The referee writes:

How practical is the TTI method in absolute terms, i.e. what is the runtime? If the Toulouse model example with χ = 50 is any indication, constructing the IF for the SIAM with χ=160 will require 100x more time.

Our response: For semi-circle spectrum, the runtime to build the MPS-IF for t=60, \delta t=0.05, and \chi=50 is about 0.54 hour, while the runtime to build the MPS-IF for t=84, \delta t=0.14, \chi=160 is 49.5 hours in our implementation, which is indeed much slower (90x times slower). This is also the reason why we stop at t = 84 for the transport problem.

The referee writes:

The paper mentions the Fishman-White algorithm, which constructs the Gaussian state of the IF by applying local gates to the vacuum state (a method to construct the partial IF). In this algorithm, the number of tensor network operations also scales roughly linearly with time. Does this imply that there is a way to construct the partial IF with similar efficiency scaling as the TTI method, even without leveraging the time translational invariance of the hybridization terms?

Our response: We thank the referee for this comment. The Fishman-White algorithm is intended for building Gaussian states, which is used in the tensor network IF algorithm. In our case we work with coherent state in terms of Grassmann variables. We believe this algorithm can still be used but it is to be explored. In principle, this algorithm, if it can be applied, can be as efficient as the TTI-IF algorithm, which is something that may be explored in future studies. However, the Fishman-White algorithm breaks the TTI property, thus can not be directly generalized to work with infinite MPSs.

The referee writes:

For all the simulations, please specify whether you are computing the IF up to time tmax and extracting observable values for all t<=tmax, or whether you are computing observable values at time t using the IF defined to time t.

Our response: We thank for the referee for the comment. We compute the IF up to the time t_max and extract observables for all t<=t_max. We have now added a last sentence in the end of Sec II.B to state this point.

The referee writes:

Regarding Figure 4, can the authors comment on why the earlier times (t=4) are less accurate than longer times when compared to exact diagonalization results?

Our response: We thank for the referee for the comment. This is a fact that we have observed in all our numerical simulations for the transient dynamics including our other works (which is opposite to the wave-function based MPS methods), but we donot fully understand it yet. In principle the MPS-IF is similar to a spatially one-dimensional many-body quantum state, for which the error should spread out in the whole time window, and the errors in different locations are expected to be similar. We guess this may be due to the first-order discretization scheme we used, which may be more affected by the initial condition.

The referee writes:

Fig 3b at time Dt/2<5 shows that improving the prony fitting error does not necessarily lead to similar order of magnitude improvements in the dynamics. Can the authors comment on why this is the case for the Lorentzian coupling strength function as opposed to the semi-circular one? Is the latter easier due to the finite support over ω? Does the error in the retarded Green's function reflect the deviation of the Prony fit?

Our response: We thank for the referee for the comment. As mentioned in the previous response, the error is spread out in the whole time window. Thus improving the prony fitting would reduce the total error of the whole time window. While the behavior of the distribution of the local errors, similar to our previous answer, is still not well understood.

The referee writes:

In Section II.B, is the statement "Based on the ADT, one can easily calculate any multi-time correlations of the impurity" accurate? Following the definition of the ADT from QUAPI and TEMPO, the ADT should already factor in system propagations as well. That is, the ADT is entire integrand of eq (3), as opposed to the IF which is only the Iσ part of the integrand. Once the system propagation tensors K are included, is it possible to freely calculate any multi-time correlations?

Our response: We thank for the referee for the comment. Yes, this statement is accurate. In TEMPO, K and I are tensors of system quantum numbers $s$, and when K is already included in the ADT it would be not possible to calculate the expectation of non-diagonal system operators (essentially, this is because that the ADT in TEMPO for the spin-boson model is only a simplified version of the full process tensor, see PRA 97. 012127 for definition of process tensor, due to the special form of coupling between impurity and bath). However, in GTEMPO, both $\bar{a}$ and $a$ are kept in track in the ADT (e. g., the ADT is the full process tensor), which enables us to calculate any correlations involving $\hat{a}^{\dagger}$ and $\hat{a}$, see Eq.(10) in Ref. 41 as an example.

In addition, in TEMPO we can also keep tracks of $s$ and its conjugate, where the conjugate of $s$ is just itself so it is usually merged to $s$ in the formalism. Then we can also calculate any correlations from the ADT with the cost that the ranks of K and I are doubled.

The referee writes:

Related to the previous point, the symbol K in eq (3) is not defined in any of the text surrounding the equation.

Our response: We thank for the referee for the comment. We have added the definition after Eq. (3).

The referee writes:

It would be useful to include an additional appendix A with the explicit expressions of the hybridization terms, as this would make the paper much more self-contained.

Our response: We thank for the referee for the comment, and we have added App. D to give corresponding explicit expressions.

The referee writes:

Is there a technical reason why the authors only consider the IF up to first order error in the time discretization?

Our response: Yes. With the same bond dimension, we found that the accuracy of the first and second order methods eventually become similar, as shown in Ref. 41. However, compared to the second order method, the first order method is preferred since it is much easier to implement.

The referee writes:

In the sentence below eq (9), it seems to imply that the Prony algorithm finds the optimal parameters for the exponential fit. Is this actually true?

Our response: The more accurate statement is: the Prony algorithm finds the the optimal parameters to fit a given function with the sum of n exponential functions. In addition, as the lambdas can be complex, those exponential can oscillate and in principle any function can be approximated by such fit, given a large enough n.

---

## Round 3 · Author Response

We thank the editors and referee for the reports. We have revised our manuscript according to the referee's comments, and the reply to the report is given in the submission page. We hope that we have significantly improved the manuscript and answered the referee's questions properly.

---

## Round 3 · List of Changes

1. Added some descriptive sentences according to the referee's comments
  2. Added an appendix about multiplication of GMPS
  3. Added some discussion about the relation between the Prony method and linear prediction method

---

## Round 4 · Referee Report · Anonymous (Referee 1) · 2024-8-14

Report

Dear SciPost Team,

thank you for forwarding the 3rd revised manuscript by Guo et al. on the Efficient construction of the Feynman-Vernon influence functional as matrix product states (2402.14350v4).

The authors addressed all concerns up to the following minor suggestions. Since at the same time the paper was transferred to SciPost Physics Core, I thus recommend publication.

Remaining comments:

The newly added Fig. 4 is very useful, but immediately raises further questions since the computed OSSE is extremely small! (Is the small OSEE a generic feature also away from Toulouse?)

Values OSEE < 0.005 suggest that the underlying operator is nearly a product operator. A bond dimension of chi=100 may be much of an overkill for this.

From the presented data one may speculate that OSEE->0 for dt->0; one may compare this to a Trotter MPO for exp(-iH*dt) which also becomes the identity product operator for dt->0; is this a sensible analogy?

A crucial extra piece of information therefore is the SVD spectrum that underlies the OSEE: how quickly does it decay? if it decays strongly, a much smaller chi may have sufficed for the same accuracy.

The inset to Fig. 4 shows `absolute error', of what quantity?

It appears that the smaller error for smaller dt is a systematic time-descretization error that is already completely converged for given chi.

following (B3-B4)

where the upper indices i_k,j_k are actual powers. this is misleading now, since this does not concern the tensor A.

Recommendation

Publish (meets expectations and criteria for this Journal)

  • validity: -
  • significance: -
  • originality: -
  • clarity: -
  • formatting: -
  • grammar: -

Author:  Ruofan Chen  on 2024-09-10  [id 4754]

(in reply to Report 1 on 2024-08-14)

We thank the referee for the refereeing and comments. Our response to the comments are shown below.

The referee writes:

The newly added Fig. 4 is very useful, but immediately raises further questions since the computed OSSE is extremely small! (Is the small OSEE a generic feature also away from Toulouse?) Values OSEE < 0.005 suggest that the underlying operator is nearly a product operator. A bond dimension of chi=100 may be much of an overkill for this. From the presented data one may speculate that OSEE->0 for dt->0; one may compare this to a Trotter MPO for exp(-iHdt) which also becomes the identity product operator for dt->0; is this a sensible analogy? A crucial extra piece of information therefore is the SVD spectrum that underlies the OSEE: how quickly does it decay? if it decays strongly, a much smaller chi may have sufficed for the same accuracy.

Our response: We thank the referee for the comment. Yes the OSEE is extremely small, and this is a generic feature determined solely by the hybridization function, independent the bare impurity Hamiltonian. The small OSEE also means that the entanglement spectrum of the MPS-IF decays very fast. And indeed bond dimension 100 is an overkill, which is only used for the purpose of Fig.4 to illustrate the growth of OSEE. For most simulations in this work, bond dimension 50 is already more than enough (for the similation of the transport problem, we have used a larger dt, and in this case a larger bond dimension 160 is used). A rough explanation of this is that the MPS-IF is similar to a high-temperature thermal state if we take F as a long-range Hamiltonian, but a more quantitive theoretical explanation is currently in lack. The fact that has been observed also in the bosonic case is that for larger dt one needs a larger bond dimension, we guess the referee could be right that OSEE->0 for dt->0. We would also like to stress that the small singular values of the MPS-IF are very important in our practice, as if we throw away singular values with a moderate threshold 10^-6, we can see that the accuracy can be affacted, and this is the reason that we essentially only use the bond dimension to compress the MPS.

The referee writes:

The inset to Fig. 4 shows `absolute error', of what quantity?

Our response: It is the absolute error of retarded Green's function, as shown in Fig. 3. We would add some sentence in the final version to describe it.

The referee writes:

following (B3-B4) where the upper indices i_k,j_k are actual powers. this is misleading now, since this does not concern the tensor A.

Our response: Thanks for the reminder. We would change it to

where the upper indices i_k, j_k of GV \xi_k are actual powers.

in the final version.

---

## Round 4 · Referee Report · Anonymous (Referee 2) · 2024-8-22

Report

I thank the authors for their thorough responses and revisions. They have noticeably amended their discussion of their method in terms of entanglement content of the MPS, along with adding a discussion contrasting a memory truncation in the spirit of QUAPI/TEMPO against their zipup algorithm. These additions, alongside minor modifications, have clarified some of my previous remarks.

A few more remarks: 1- Is eq (4) correct as written? I would have expected there to be $\bar{a}$ GVs. 2- I understand that the memory truncation as presented in Appendix D is incompatible with the zipup algorithm. My suggestion of a memory truncation and its potential to improve computational costs was aimed at the level of the IF, not the ADT. I imagine, for example, that the partial IF (eq (8)) would be truncated to $\Delta k_{max}$ terms. Each of these terms would be multiplied into the IF as it is being constructed. As a further approximation to the MPS compression, singular value truncations would occur only over those $\Delta k_{max}$ sites. This would be similar in spirit to Jorgenson and Pollock's construction in Phys. Rev. Lett. 123, 240602 (2019). This would avoid the problem pointed out by the authors that the ADT would need to be explicitly constructed. In my mind this would have been a better comparison to the authors' TTI IF method, as far as computational cost is concerned. 3- While it is nice to have information on the OSEE, what would be better is if the figure showed either how the OSEE in the MPS changes with the max bond dimension chi, or if it showed the entanglement spectrum explicitly.

As these are only minor comments, I am happy to recommend publication of this manuscript in SciPost Phys Core.

Recommendation

Publish (meets expectations and criteria for this Journal)

  • validity: -
  • significance: -
  • originality: -
  • clarity: -
  • formatting: -
  • grammar: -

Author:  Ruofan Chen  on 2024-09-10  [id 4755]

(in reply to Report 2 on 2024-08-22)

We thank the referee for the refereeing and comments. Our response to the comments are shown below.

The referee writes:

Is eq (4) correct as written? I would have expected there to be \bar{a} GVs.

Our response: Yes, it is correct. The ket \ket{a} indicates \bar{a}, so usually the bar is not shown explicitly.

The referee writes:

I understand that the memory truncation as presented in Appendix D is incompatible with the zipup algorithm. My suggestion of a memory truncation and its potential to improve computational costs was aimed at the level of the IF, not the ADT. I imagine, for example, that the partial IF (eq (8)) would be truncated to Δkmax terms. Each of these terms would be multiplied into the IF as it is being constructed. As a further approximation to the MPS compression, singular value truncations would occur only over those Δkmax sites. This would be similar in spirit to Jorgenson and Pollock's construction in Phys. Rev. Lett. 123, 240602 (2019). This would avoid the problem pointed out by the authors that the ADT would need to be explicitly constructed. In my mind this would have been a better comparison to the authors' TTI IF method, as far as computational cost is concerned.

Our response: Thanks for the suggestions, and we have considered this kind of memory truncation earlier. However, since the most expensive part is due to the large bond dimension and global SVD truncation is in principle more numerical stable than truncation within Δkmax terms, we decide not to employ such a memory truncation at this time to reduce the source of errors. The memory truncation in GTEMPO may be explored in the future.

The referee writes:

While it is nice to have information on the OSEE, what would be better is if the figure showed either how the OSEE in the MPS changes with the max bond dimension chi, or if it showed the entanglement spectrum explicitly.

Our response: Thanks for the suggestion. However, we have already used a large bond dimension, which is already more than necessary, in the figure to avoid the inaccuracy of OSEE. Therefore the OSEE plotted is expected to be accurate enough, and an additional figure is not necessary.

---

## Round 4 · List of Changes

1.Added some descriptive sentences according to the referee's comments
2. Added an appendix about explicit QUAPI expressions
3. Added a figure about OSEE

---

## Editorial Decision

published